# Sparser Block-Sparse Attention via Token Permutation

## Abstract

Scaling the context length of large language models (LLMs) offers significant benefits but is computationally expensive. This expense stems primarily from the self-attention mechanism, whose $O(N^2)$ complexity with respect to sequence length presents a major bottleneck for both memory and latency. Fortunately, the attention matrix is often sparse, particularly for long sequences, suggesting an opportunity for optimization. Block-sparse attention has emerged as a promising solution that partitions sequences into blocks and skips computation for a subset of these blocks. However, the effectiveness of this method is highly dependent on the underlying attention patterns, which can lead to sub-optimal block-level sparsity. For instance, important key tokens for queries within a single block may be scattered across numerous other blocks, leading to computational redundancy. In this work, we propose Permuted Block-Sparse Attention (**PBS-Attn**), a plug-and-play method that leverages the permutation properties of attention to increase block-level sparsity and enhance the computational efficiency of LLM prefilling. We conduct comprehensive experiments on challenging long-context datasets, demonstrating that PBS-Attn consistently outperforms existing block-sparse attention methods in model accuracy and closely matches the full attention baseline. Powered by our custom permuted-FlashAttention kernels, PBS-Attn achieves an end-to-end speedup of up to $2.75\times$ in long-context prefilling, confirming its practical viability. Code will be released after the reviewing period.

## 1 Introduction

Modern Large Language Models (LLMs) have demonstrated remarkable proficiency in handling long-context tasks (OpenAI, 2025; Gemini Team, Google, 2025; Anthropic, 2025), a capability fueled by advancements in infrastructure (Liu et al., 2023; Jin et al., 2024), training methodologies (Yang et al., 2025a), and novel positional embedding schemes (Su et al., 2023; Press et al., 2022; Peng et al., 2023). This progress enables models to process context windows spanning thousands or even millions of tokens, unlocking novel applications such as analyzing entire codebases, summarizing lengthy legal documents, and interpreting long-form video content.

However, this extended capability is constrained by prohibitive memory and computational overheads. This bottleneck primarily stems from the self-attention mechanism within the Transformer architecture (Vaswani et al., 2023). The necessity for each token to attend to all other tokens results in a computational complexity that scales quadratically with the input sequence length, posing a fundamental challenge to scalable and accessible long-context processing.

To address this challenge, researchers have proposed solutions from multiple perspectives. Architecturally, some approaches replace the standard quadratic attention with sub-quadratic alternatives, such as linear transformers (Katharopoulos et al., 2020; Yang et al., 2025d). Others substitute the attention mechanism entirely with alternatives like State Space Models (SSMs), which operate recurrently to process extremely long sequences with high efficiency (Gu & Dao, 2024; Dao & Gu, 2024; Yang et al., 2025c). Concurrently, hardware-aware optimizations, exemplified by FlashAttention (Dao et al., 2022), reduce memory overhead by tiling the sequence into blocks and performing an online softmax computation. This method avoids the materialization of the full attention matrix, thereby alleviating the memory overhead and efficiency constraints imposed by I/O limitations. Building directly upon this tiled approach, block-sparse attention further reduces computation by

skipping the computation for certain blocks using a pre-computed sparse block mask (Dao et al., 2022; Jiang et al., 2024; Lai et al., 2025; Xu et al., 2025; Zhang et al., 2025; Gao et al., 2025). This technique leverages the inherent sparsity of attention matrices, wherein most of the attention mass for a given query is concentrated on a small subset of key tokens. This property, particularly prominent in long sequences, allows for a drastic reduction in computation without significantly compromising performance. While this block-level approach maximizes parallel efficiency, its rigidity can lead to a sub-optimal sparsity pattern. This issue arises when the key tokens relevant to queries within a single block are widely scattered, collectively spanning an unnecessarily large number of key blocks and thereby forcing redundant computation.

Fortunately, the same token-wise computation that leads to quadratic complexity also presents an opportunity to mitigate it. The attention mechanism is permutation-invariant, meaning we can reorder the query and key sequences to achieve a more favorable block-sparse structure and further improve block sparsity. Leveraging this insight, we propose **Permuted Block-Sparse Attention (PBS-Attn)**, a plug-and-play strategy that reorganizes query and key sequences to accelerate LLM prefilling. To accommodate causal attention for LLMs, we introduce a novel segmented permutation strategy that preserves inter-segment causality while applying intra-segment permutation. Extensive experiments demonstrate that PBS-Attn increases block-level sparsity, yielding significant efficiency gains with minimal degradation in model performance. Specifically, powered by our custom permuted-FlashAttention kernels, PBS-Attn achieves an end-to-end speedup of up to $2.75\times$ in LLM prefilling, while maintaining performance close to the full attention baseline ondatasets like LongBench (Bai et al., 2024), LongBenchv2 (Bai et al., 2025) and RULER (Hsieh et al., 2024).

## 2 PRELIMINARIES

**Scaled Dot-Product Attention**    As the cornerstone of modern large language models, the attention mechanism facilitates a dynamic synthesis of information by calculating a weighted aggregation of value ($\boldsymbol{V}$) vectors. These weights, or attention scores, are determined by the dot-product similarity between a given token's query ($\boldsymbol{Q}$) vector and the key ($\boldsymbol{K}$) vectors of all other tokens in the sequence. This process allows the model to directly assess the relevance of every token relative to every other, enabling the effective capture of long-range dependencies, but at a cost of quadratic complexity over the sequence length. Formally, the attention mechanism is given by:

$$\boldsymbol{A} = \text{softmax}\left(\frac{\boldsymbol{Q}\boldsymbol{K}^T}{\sqrt{d}}\right) \tag{1}$$

$$\text{Attention}(\boldsymbol{Q}, \boldsymbol{K}, \boldsymbol{V}) = \boldsymbol{A}\boldsymbol{V} \tag{2}$$

where $d$ is the head dimension for multi-head attention and $\boldsymbol{A}$ is the attention matrix.

**FlashAttention**    FlashAttention (Dao et al., 2022) employs a tiled approach that partitions the input sequence into blocks and performs an online softmax computation. This strategy circumvents the materialization of the full attention matrix $\boldsymbol{A}$, which significantly reduces memory overhead and improves efficiency for I/O-bound operations on GPUs.

Formally, let the input query, key and value matrices be $\boldsymbol{Q} \in \mathbb{R}^{N \times d}$, $\boldsymbol{K} \in \mathbb{R}^{M \times d}$, and $\boldsymbol{V} \in \mathbb{R}^{M \times d}$ and divide them into $T_r = \lceil \frac{N}{B} \rceil$ and $T_c = \lceil \frac{M}{B} \rceil$ blocks with block size $B$(we use the same block size for $\boldsymbol{Q}$ and $\boldsymbol{K}/\boldsymbol{V}$ for simple terminology), $\boldsymbol{Q} = [\boldsymbol{Q}_1, \ldots, \boldsymbol{Q}_{T_r}]$, $\boldsymbol{K} = [\boldsymbol{K}_1, \ldots, \boldsymbol{K}_{T_c}]$, and $\boldsymbol{V} = [\boldsymbol{V}_1, \ldots, \boldsymbol{V}_{T_c}]$. For query block $\boldsymbol{Q}_i$, the computation for the corresponding output block $\boldsymbol{O}_i$ is defined by a system of recursive equations over the key/value blocks $j = 1, \ldots, T_c$. The state at step $j$ is the triplet $(\boldsymbol{O}_i^{(j)}, \boldsymbol{m}_i^{(j)}, \boldsymbol{l}_i^{(j)})$. The state is initialized at $j = 0$ with $\boldsymbol{O}_i^{(0)} = \boldsymbol{0}$, $\boldsymbol{m}_i^{(0)} = -\infty$, and $\boldsymbol{l}_i^{(0)} = \boldsymbol{0}$. For each step $j = 1, \ldots, T_c$, given the intermediate scores $\boldsymbol{S}_{ij} = \frac{\boldsymbol{Q}_i \boldsymbol{K}_j^T}{\sqrt{d}}$ and local maximum $\boldsymbol{m}'_{ij} = \text{row\_max}(\boldsymbol{S}_{ij})$, the state is updated from $j - 1$ to $j$:

$$\boldsymbol{m}_i^{(j)} = \max(\boldsymbol{m}_i^{(j-1)}, \boldsymbol{m}'_{ij}) \tag{3}$$

$$\boldsymbol{l}_i^{(j)} = \boldsymbol{l}_i^{(j-1)} e^{\boldsymbol{m}_i^{(j-1)} - \boldsymbol{m}_i^{(j)}} + \text{row\_sum}(\exp(\boldsymbol{S}_{ij} - \boldsymbol{m}_i^{(j)})) \tag{4}$$

$$\boldsymbol{O}_i^{(j)} = \boldsymbol{O}_i^{(j-1)} e^{\boldsymbol{m}_i^{(j-1)} - \boldsymbol{m}_i^{(j)}} + \exp(\boldsymbol{S}_{ij} - \boldsymbol{m}_i^{(j)})\boldsymbol{V}_j \tag{5}$$

After the final step, the output is normalized as $\boldsymbol{O}_i = \text{diag}\left((\boldsymbol{l}_i^{(T_c)})^{-1}\right)\boldsymbol{O}_i^{(T_c)}$.

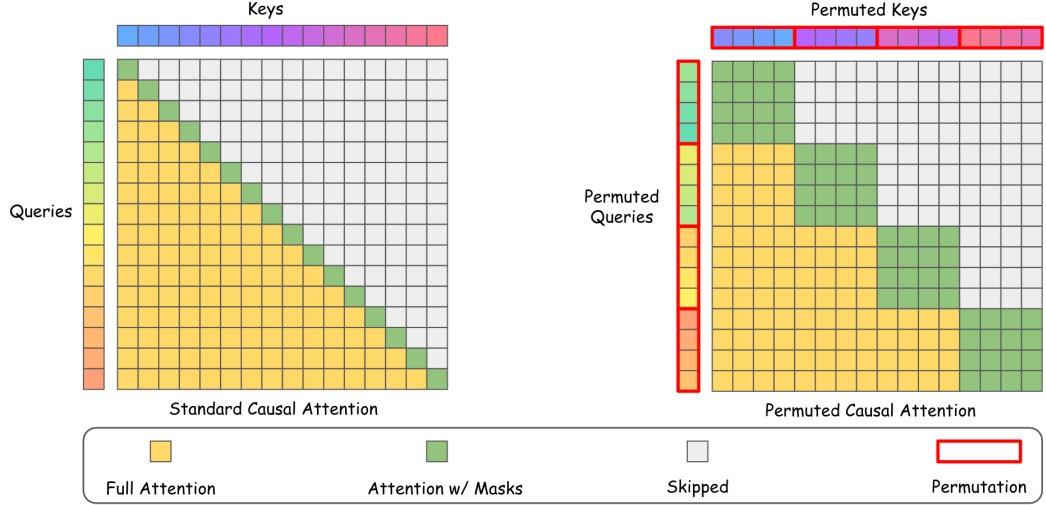

Figure 1: Illustration of causal attention without (**Left**) and with (**Right**) segmented permutation with $B = 1, S = 4$. Segmented permutation enhances block-level sparsity via intra-segment permutation while preserving inter-segment causality. By restricting computation of blocks within on-diagonal segments (green blocks), we can safely skip inter-segment blocks (yellow blocks) for block-sparse attention.

**Block-Sparse Attention** Building upon the tiled computation of FlashAttention, block-sparse attention introduces a further layer of optimization by selectively pruning block-wise interactions. This is achieved using a predefined sparse block mask, $\boldsymbol{M} \in \{0, 1\}^{T_r \times T_c}$. For any given query block $\boldsymbol{Q}_i$, the attention computation is only performed against key-value blocks $\boldsymbol{K}_j$ and $\boldsymbol{V}_j$ where the corresponding mask entry $\boldsymbol{M}_{ij} = 1$.

If $\boldsymbol{M}_{ij} = 0$, the calculation of the score matrix $\boldsymbol{S}_{ij}$ and the subsequent state update steps are entirely bypassed. Consequently, the state remains unchanged from the previous iteration; that is, $(\boldsymbol{O}_i^{(j)}, \boldsymbol{m}_i^{(j)}, \boldsymbol{l}_i^{(j)}) = (\boldsymbol{O}_i^{(j-1)}, \boldsymbol{m}_i^{(j-1)}, \boldsymbol{l}_i^{(j-1)})$.

## 3 PERMUTED BLOCK-SPARSE ATTENTION

### 3.1 PERMUTATION PROPERTIES OF ATTENTION

The attention mechanism exhibits specific symmetries with respect to permutations of its inputs, which we formalize in the following lemmas.

**Lemma 3.1** (Key-Value Pair Permutation Invariance). *The attention mechanism is invariant to the order of the source sequence, provided that the key-value pairings are maintained.*

*Formally, let $\boldsymbol{P}_\pi \in \{0, 1\}^{M \times M}$ be a permutation matrix that reorders the rows of a matrix according to a permutation $\pi$ on the index set $\{1, \ldots, M\}$. The following identity holds:*

$$\text{Attention}(\boldsymbol{Q}, \boldsymbol{P}_\pi \boldsymbol{K}, \boldsymbol{P}_\pi \boldsymbol{V}) = \text{Attention}(\boldsymbol{Q}, \boldsymbol{K}, \boldsymbol{V}) \tag{6}$$

**Lemma 3.2** (Query Permutation Equivariance). *The attention mechanism is equivariant with respect to permutations of the query sequence.*

*Formally, let $\boldsymbol{P}_\sigma \in \{0, 1\}^{N \times N}$ be a permutation matrix that reorders the rows of a matrix according to a permutation $\sigma$ on the index set $\{1, \ldots, N\}$. The following relationship holds:*

$$\text{Attention}(\boldsymbol{P}_\sigma \boldsymbol{Q}, \boldsymbol{K}, \boldsymbol{V}) = \boldsymbol{P}_\sigma \text{Attention}(\boldsymbol{Q}, \boldsymbol{K}, \boldsymbol{V}) \tag{7}$$

The proofs of Lemma 3.1 and 3.2 are provided in Appendix A.1 and A.2, respectively.

Combining these properties, we arrive at a general theorem for attention under simultaneous input permutations. A detailed proof is provided in Appendix A.3.

**Theorem 3.3** (Attention Permutation Invariance under Inverse Transformation). *If the queries are permuted by $\boldsymbol{P}_\sigma$ and the key-value pairs are permuted by $\boldsymbol{P}_\pi$, the resulting output is a permuted*

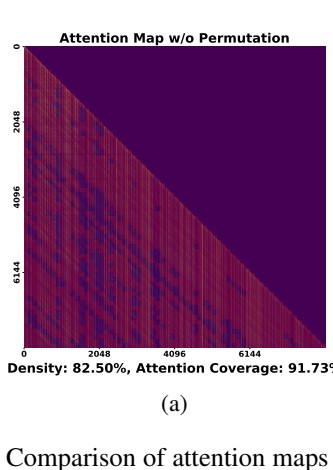
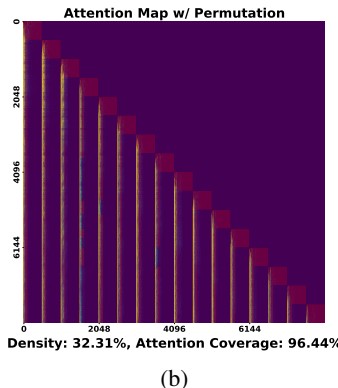

**Attention Map w/o Permutation**

Density: 82.50%, Attention Coverage: 91.73%

(a)

**Attention Map w/ Permutation**

Density: 32.31%, Attention Coverage: 96.44%

(b)

Figure 2: Comparison of attention maps for Llama-3.1-8B (layer 0, head 30) on an 8K LongBench example, showing the pattern without (a) and with (b) segmented permutation. The red overlay indicates blocks selected for block-sparse attention, and the attention coverage is calculated as the total attention scores covered by the selected blocks. More visualizations are provided in Appendix G.

*version of the original output. Applying the inverse of the query permutation recovers the original, unpermuted output. Formally:*

$$P_\sigma^T \, Attention(P_\sigma Q, P_\pi K, P_\pi V) = Attention(Q, K, V) \tag{8}$$

Theorem 3.3 establishes that the query matrix $Q$ and key matrix $K$ can be permuted by $P_\sigma$ and $P_\pi$ respectively, provided that $P_\pi$ is also applied to the value matrix $V$ and $P_\sigma^T$ to the output $O'$. This property enables the rearrangement of the attention matrix $A$, without affecting the attention output.

## 3.2 SEGMENTED PERMUTATION FOR CAUSAL ATTENTION

Motivated by Theorem 3.3, we explore whether its permutation properties can be leveraged to restructure the attention matrix. This rearrangement promises higher block sparsity and computational savings, particularly during the compute-bound prefill stage of inference. The primary objective is to **co-locate the salient key tokens corresponding to queries from the same computational block, thereby enhancing block-level sparsity.**

However, a critical challenge remains: maintaining causality post-permutation. Specifically, LLMs are trained with causal attention, which restricts queries to attending only to keys in preceding positions, resulting in a lower-triangular attention matrix, $A$. During prefilling, blocks above the main diagonal are computationally redundant and can be skipped; consequently, the original block density for causal attention is $\frac{T_c+1}{2T_c}$. A naive application of a global permutation to the query and key sequences would dismantle this vital causal structure. Such a permutation could scatter dependencies across the entire matrix, potentially transforming the sparse, lower-triangular structure into a fully dense one (i.e., a block density of 1).

To address this challenge, we propose a segmented permutation strategy that preserves *inter-segment causality* while applying *intra-segment permutation*, illustrated in Figure 1. Formally, we partition the initial $\lfloor N/S \rfloor \cdot S$ tokens of the input sequences $Q, K, V$ into $G = \lfloor N/S \rfloor$ non-overlapping, contiguous segments of size $S$. The remaining $N \pmod{S}$ tokens are left unpermuted.

Let $Q_i, K_i, V_i \in \mathbb{R}^{S \times d}$ denote the $i$-th segment for $i \in \{1, \ldots, G\}$. For each segment $i$, we introduce local permutations, $\sigma_i$ for queries and $\pi_i$ for keys, that reorder tokens within that segment. The global permutation operators, $P_\sigma$ and $P_\pi$, are then constructed as block-diagonal matrices from these respective local permutations. For the key permutation matrix $P_\pi$:

$$P_\pi = \text{diag}(P_{\pi_1}, \ldots, P_{\pi_G}, I_{N \pmod S}) = \begin{pmatrix} P_{\pi_1} & 0 & \cdots & 0 & 0 \\ 0 & P_{\pi_2} & \cdots & 0 & 0 \\ \vdots & \vdots & \ddots & \vdots & \vdots \\ 0 & 0 & \cdots & P_{\pi_G} & 0 \\ 0 & 0 & \cdots & 0 & I_{N \pmod S} \end{pmatrix} \tag{9}$$

Here, each $\boldsymbol{P}_{\pi_i} \in \{0,1\}^{S \times S}$ is the permutation matrix for the local key permutation $\pi_i$, and $\boldsymbol{I}_{N \pmod S}$ is the identity matrix corresponding to the last incomplete segment. The query permutation matrix $\boldsymbol{P}_\sigma$ is constructed analogously from its own set of local permutations, $\{\sigma_i\}_{i=1}^{G}$.

## 3.3 QUERY-AWARE KEY PERMUTATION

The inherent sparsity of the attention mechanism implies that for any given query, a small subset of key tokens accounts for most of the attention mass. A prominent pattern is that certain keys are consistently important across almost all queries. This phenomenon is widely recognized in the literature as "heavy hitter" keys (Zhang et al., 2023) or "Vertical Lines" in the attention map (Jiang et al., 2024; Lai et al., 2025).

To maintain model accuracy, block-sparse attention must capture all "heavy hitter" keys. However, this constraint can severely limit block sparsity when these key tokens are scattered throughout the sequence, spanning a large number of blocks, as shown in Figure 2a.

To mitigate this, we propose a query-aware key permutation strategy that effectively identifies and clusters "heavy hitter" keys, thereby significantly improving block-level sparsity. This method implements the permutation as an efficient, segment-wise sorting process. Within each segment, keys are sorted based on their estimated average attention scores using the last block of queries. Concretely, we first compute a global importance score vector $\boldsymbol{s} \in \mathbb{R}^N$ for all keys in the sequence using the last block of queries, $\boldsymbol{Q}_{\text{last\_block}}$:

$$\boldsymbol{s} = \text{mean}_{\text{rows}} \left( \text{softmax} \left( \frac{\boldsymbol{Q}_{\text{last\_block}} \boldsymbol{K}^T}{\sqrt{d}} \right) \right) \tag{10}$$

The local permutation $\pi_i$ for each segment $i$ is then obtained by sorting the keys within that segment based on $\boldsymbol{s}$ in descending order:

$$\pi_i = \text{argsort}(-\boldsymbol{s}_{[(i-1)S+1:iS]}) \tag{11}$$

As shown in Figure 2b, this permutation strategy can effectively cluster the "Vertical Lines" thereby significantly improving block-level sparsity while maintaining attention coverage.

To further validate the effectiveness of permutation, we conduct a comprehensive analysis in Appendix B. There, we formally define block-level attention coverage and perform statistical evaluations to quantify the sparsity gains from permutation across three axes: method-wise, layer-wise, and head-wise. **Method-wise**, results show that sparsity gains primarily stem from clustering "heavy hitter" keys rather than utilizing local query centroids. Furthermore, permutation effectiveness is shown to be insensitive to the specific subset of queries used for scoring. **Layer-wise**, sparsity improvement increases with the coverage level for most layers, with middle-to-deep layers benefiting the most. **Head-wise**, attention heads respond differently to permutation. While the majority show significant or moderate sparsity gains (Figure 2), a minority exhibit marginal gains or even degradation (Figure 9). We further provide a failure mode analysis for these cases in Appendix B.

## 3.4 PERMUTED BLOCK-SPARSE ATTENTION

The proposed permuted block-sparse attention (**PBS-Attn**) mechanism is detailed in Algorithm 1, where the key adjustments relative to FlashAttention are highlighted in red. The process commences by permuting the query, key, and value matrices. Specifically, we apply permutation $P_\sigma$ to the query matrix $\boldsymbol{Q}$ and $P_\pi$ to the key matrix $\boldsymbol{K}$, while the value matrix $\boldsymbol{V}$ shares the same permutation $P_\pi$, as justified by Lemma 3.1. Subsequently, a block selection algorithm is applied to the permuted queries and keys, yielding a block-sparse mask $M$. This mask, $\boldsymbol{M}$, governs the tiled attention computation by dictating which block-wise operations can be pruned. For selected blocks (where $M_{i,j} = 1$), a standard online softmax attention is computed, updating the state of the permuted output block $\boldsymbol{O}'_i$. For unselected blocks (where $M_{i,j} = 0$), this computation is skipped, and the state of $\boldsymbol{O}'_i$ remains unchanged. For the block selection algorithm, we use a simple strategy that utilizes mean pooling and block-wise attention to estimate the importance of each key block for each query block for the main method, where we detail in Appendix C.1. Crucially, we demonstrate that the sparsity improvements conferred by permutation are agnostic to the specific block selection algorithm, allowing our method to be combined with existing algorithms to further improve block

---

**Algorithm 1** Permuted Block-Sparse Attention

---

**Require:** $\boldsymbol{Q}, \boldsymbol{K}, \boldsymbol{V} \in \mathbb{R}^{N \times d}$, permutation matrices $\boldsymbol{P}_\sigma, \boldsymbol{P}_\pi \in \{0,1\}^{N \times N}$, segment size $S$, block size $B$

**Ensure:** Permuted attention output $\boldsymbol{O} \in \mathbb{R}^{N \times d}$

    $\boldsymbol{Q}' \leftarrow \boldsymbol{P}_\sigma \boldsymbol{Q}, \boldsymbol{K}' \leftarrow \boldsymbol{P}_\pi \boldsymbol{K}, \boldsymbol{V}' \leftarrow \boldsymbol{P}_\pi \boldsymbol{V}$          ▷ Apply permutation

    Divide $\boldsymbol{Q}'$ into $T_r = \lceil \frac{N}{B} \rceil$ blocks $\boldsymbol{Q}_1', \ldots, \boldsymbol{Q}_{T_r}'$; divide $\boldsymbol{K}', \boldsymbol{V}'$ into $T_c = \lceil \frac{N}{B} \rceil$ blocks $\boldsymbol{K}_1', \ldots, \boldsymbol{K}_{T_c}'$ and $\boldsymbol{V}_1', \ldots, \boldsymbol{V}_{T_c}'$;

    $M \leftarrow \text{BLOCK\_SELECTION}(\boldsymbol{Q}', \boldsymbol{K}', B, S)$          ▷ Select blocks, see Appendix C

    Initialize $\boldsymbol{O}' \leftarrow \boldsymbol{0}$;

    **for** $i = 1$ to $T_r$ **do**

        Load $\boldsymbol{Q}_i'$ to SRAM; Initialize $\boldsymbol{O}_i^{(0)} \leftarrow \boldsymbol{0}, \boldsymbol{m}_i^{(0)} \leftarrow -\infty, \boldsymbol{l}_i^{(0)} \leftarrow \boldsymbol{0}$;

        **for** $j = 1$ to $T_c$ **do**

            **if** $M_{i,j} = 1$ **then**          ▷ Compute attention only for selected blocks

                Load $\boldsymbol{K}_j', \boldsymbol{V}_j'$ to SRAM;

                Compute $\boldsymbol{S}_{ij}' = \boldsymbol{Q}_i' \boldsymbol{K}_j'^T / \sqrt{d}, \boldsymbol{m}_i^{(j)} = \max(\boldsymbol{m}_i^{(j-1)}, \text{row\_max}(\boldsymbol{S}_{ij}'))$;

                Compute $\boldsymbol{l}_i^{(j)} = \boldsymbol{l}_i^{(j-1)} e^{\boldsymbol{m}_i^{(j-1)} - \boldsymbol{m}_i^{(j)}} + \text{row\_sum}(\exp(\boldsymbol{S}_{ij}' - \boldsymbol{m}_i^{(j)}))$;

                Compute $\boldsymbol{O}_i^{(j)} = \boldsymbol{O}_i^{(j-1)} e^{\boldsymbol{m}_i^{(j-1)} - \boldsymbol{m}_i^{(j)}} + \exp(\boldsymbol{S}_{ij}' - \boldsymbol{m}_i^{(j)}) \boldsymbol{V}_j'$;

           **else**

                $\boldsymbol{O}_i^{(j)} \leftarrow \boldsymbol{O}_i^{(j-1)}, \boldsymbol{m}_i^{(j)} \leftarrow \boldsymbol{m}_i^{(j-1)}, \boldsymbol{l}_i^{(j)} \leftarrow \boldsymbol{l}_i^{(j-1)}$;          ▷ Skip computation

           **end if**

        **end for**

        $\boldsymbol{O}_i' \leftarrow \text{diag}((\boldsymbol{l}_i^{(T_c)})^{-1}) \boldsymbol{O}_i^{(T_c)}$; Write $\boldsymbol{O}_i'$ back to its rows in $\boldsymbol{O}'$;

    **end for**

    $\boldsymbol{O} \leftarrow \boldsymbol{P}_\sigma^T \boldsymbol{O}'$          ▷ Reverse permutation

    **return** $\boldsymbol{O}$

---

sparsity, as shown in Appendix C.2. Finally, an inverse permutation, $P_\sigma^T$, is applied to the output $\boldsymbol{O}'$ to restore the original ordering, as established by Theorem 3.3.

## 4 EXPERIMENTS

### 4.1 SETTINGS

**Models & Datasets** We employ two state-of-the-art long-context LLMs, claiming support for available context lengths above 128K tokens: **Llama-3.1-8B(128K)** (Grattafiori et al., 2024) and **Qwen-2.5-7B-1M(1M)** (Yang et al., 2025a). We evaluate the sparse attention methods on two challenging real-world long-context datasets to validate their effectiveness in real-world scenarios: **LongBench** (Bai et al., 2024) and **LongBenchv2** (Bai et al., 2025). LongBench is a collection of 21 long-context understanding tasks in 6 categories with mostly real-world data, with the average length of most tasks ranging from 5K to 15K. LongBenchv2 further scales the context length, ranging from 8K to 2M, covering various realistic scenarios. We also conduct evaluation on **RULER** (Hsieh et al., 2024), a synthetic benchmark designed to systematically evaluate long-context LLMs across various context lengths.

**Baselines** We evaluate PBS-Attn alongside a set of strong baselines to validate its effectiveness. (1) **Full Attention**: The standard attention mechanism that computes the full attention matrix as the oracle method. Specifically, we use the FlashAttention (Dao et al., 2022) implementation. (2) **Minference** (Jiang et al., 2024): A sparse attention method that performs offline attention pattern search, we utilize the official configuration for attention pattern setting. (3) **FlexPrefill** (Lai et al., 2025): A block selection method for block-sparse attention that performs block selection based on the input and selects the attention pattern on the fly. We use $\gamma = 0.95, \tau = 0.1$ as reported in the original paper. (4) **XAttention** (Xu et al., 2025): A block selection method for block-sparse attention that selects blocks based on an antidiagonal scoring of blocks. We use threshold $= 0.9, \text{stride} = 8$ as reported in the original paper. (5) **MeanPooling**: This method uses a mean pooling strategy on the unpermuted queries and keys to select blocks, which is the same selection method for PBS-

Attn(detailed in C.1). Our experiments shows that MeanPooling can serve as a strong baseline when the first and the most recent key blocks are forcibly selected for each query block, due to the attention sink phenomenon (Xiao et al., 2024). We use a selection threshold of 0.9 for MeanPooling.

**Implementation Details** For PBS-Attn, we use a block size of $B = 128$ and a segment size of $S = 256$. The block selection threshold is set to 0.9 through all experiments. We implement a custom permuted-FlashAttention kernel in Triton (Tillet et al., 2019) for efficient inference of PBS-Attn. For model inference, we replace the prefilling process with PBS-Attn or baseline methods, while keeping the decoding process as in the original attention implementation. To handle Grouped Query Attention (GQA), our default strategy replicates keys and values within each group to maximize sparsity gains. We also evaluate the feasibility of sharing the permutation within a GQA group to improve memory efficiency, as detailed in Appendix E. The experiments are conducted in a computing environment with NVIDIA H100 80GB GPUs.

Table 1: Performance comparison of various sparse attention methods on LongBench. **Bold** and underlined scores indicate the best and second-best performing methods in each category, respectively, with the exception of the full attention baseline.

| Method | Single-Doc QA | Multi-Doc QA | Summarization | Few-shot Learning | Code | Synthetic | Avg. |
|---|---|---|---|---|---|---|---|
| *Llama-3.1-8B* | | | | | | | |
| Full | 48.80 | 41.80 | 17.79 | 29.73 | 24.77 | 66.82 | 38.28 |
| MInference | 47.21 | 40.93 | 17.72 | 29.36 | 24.77 | 62.36 | 37.06 |
| FlexPrefill | 47.03 | 38.57 | 17.78 | 30.38 | 24.88 | 24.71 | 30.56 |
| XAttention | **48.26** | 40.23 | **17.86** | **31.35** | **26.19** | 54.64 | 36.42 |
| MeanPooling | 46.61 | 40.66 | 17.85 | 30.64 | 26.10 | 58.14 | 36.67 |
| **PBS-Attn** | 48.00 | **42.09** | 17.72 | 28.36 | 24.25 | **63.80** | **37.37** |
| *Qwen-2.5-7B-1M* | | | | | | | |
| Full | 44.21 | 42.97 | 16.01 | 47.48 | 3.91 | 67.50 | 37.01 |
| MInference | 42.82 | 41.76 | 16.01 | 46.41 | 3.80 | **66.50** | 36.21 |
| FlexPrefill | 38.44 | 37.51 | 15.87 | 46.12 | **6.46** | 26.67 | 28.51 |
| XAttention | **43.82** | **42.21** | 16.07 | 48.30 | 3.83 | 63.33 | 36.26 |
| MeanPooling | 39.39 | 40.96 | 15.95 | **49.07** | 4.80 | 40.83 | 31.83 |
| **PBS-Attn** | 43.01 | 41.40 | **16.12** | 47.36 | 4.00 | 66.33 | **36.37** |

## 4.2 MAIN RESULTS

**LongBench** Table 1 presents a performance comparison of various sparse attention methods on the LongBench benchmark, evaluated using the Llama-3.1-8B and Qwen-2.5-7B-1M models. As the results indicate, the unpermuted MeanPooling method already establishes a strong baseline. Crucially, by incorporating our proposed permutation strategy, PBS-Attn significantly improves performance, surpassing other block-sparse attention methods and closely approaching the performance of the oracle full-attention baseline. PBS-Attn consistently achieves the best overall performance across both models, demonstrating its effectiveness and robustness.

**LongBenchv2** To rigorously evaluate the effectiveness of PBS-Attn in extreme long-context scenarios, we conducted experiments on the more challenging LongBenchv2 benchmark. The results, presented in Table 2, reveal that PBS-Attn exhibits minimal performance degradation compared to the full attention baseline while consistently surpassing other block-sparse attention methods. Notably, PBS-Attn consistently outperforms the unpermuted MeanPooling baseline. This advantage is particularly pronounced for the Qwen-2.5-7B-1M model, where permutation brings a remarkable relative improvement of 31% in overall performance. This indicates that permutation can successfully group key tokens that have similar behaviors, making the block selection more precise and covering more critical key tokens.

**RULER** To systematically evaluate PBS-Attn across various lengths, we conduct experiments on the RULER benchmark, with results presented in Table 3. Due to the synthetic nature of the RULER dataset, mean pooling selection drastically diminishes performance on tasks retrieving key-value pairs in random UUIDs, necessitating the use of token-level attention in block scoring for these tasks. Therefore, we also incorporate PBS-Attn$^+$, which adopts the antidiagonal block scoring scheme proposed in XAttention (Xu et al., 2025). Notably, both PBS-Attn and PBS-Attn$^+$ consistently

Table 2: Performance comparison of various sparse attention methods on LongBenchv2. **Bold** and underlined scores indicate the best and second-best performing methods for each model, respectively, with the exception of the full attention baseline.

| Method | Llama-3.1-8B | Qwen2.5-7B-1M |
|---|---|---|
| Full | 28.83 | 35.19 |
| Minference | 29.03 | 34.19 |
| FlexPrefill | 27.24 | 27.83 |
| XAttention | 29.62 | 34.19 |
| MeanPooling | 29.42 | 26.24 |
| **PBS-Attn** | **29.82** | **34.39** |

Table 3: Results on the RULER benchmark. PBS-Attn$^+$ denotes PBS-Attn with antidiagonal scoring for block selection (Xu et al., 2025).

| Method | Llama-3.1-8B | | | | | | | Qwen-2.5-7B-1M | | | | | | |
|---|---|---|---|---|---|---|---|---|---|---|---|---|---|---|
| | 4K | 8K | 16K | 32K | 64K | 128K | Avg | 4K | 8K | 16K | 32K | 64K | 128K | Avg |
| Full | 96.14 | 94.24 | 92.19 | 86.06 | 84.60 | 75.30 | 88.09 | 95.34 | 92.45 | 93.49 | 89.06 | 84.73 | 74.23 | 88.22 |
| Minference | **95.98** | 93.67 | **91.95** | 85.55 | **83.48** | 70.47 | 86.85 | 94.01 | 91.30 | 91.60 | 89.09 | 81.30 | 70.10 | 86.23 |
| FlexPrefill | 92.87 | 92.99 | 91.35 | 84.91 | 82.62 | 71.07 | 85.97 | 84.17 | 87.74 | 86.73 | 84.21 | 78.15 | 61.66 | 80.44 |
| XAttention | 95.63 | **93.95** | 91.63 | 86.32 | 80.54 | 70.68 | 86.46 | 93.69 | 92.10 | 91.50 | 88.35 | 81.26 | 73.05 | 86.66 |
| MeanPooling | 94.15 | 92.72 | 89.94 | 83.95 | 76.46 | 59.32 | 82.76 | 90.15 | 87.43 | 86.38 | 82.70 | 78.86 | 67.51 | 82.17 |
| **PBS-Attn** | 95.83 | **93.85** | 91.46 | 85.18 | 82.51 | 66.98 | 85.97 | 93.27 | 90.77 | 90.54 | 85.54 | 81.50 | 70.61 | 85.37 |
| **PBS-Attn$^+$** | 95.72 | 93.85 | 91.23 | **87.05** | 81.27 | **72.09** | **86.87** | **94.06** | **92.24** | **92.59** | **89.31** | **84.37** | **73.71** | **87.71** |

outperform their unpermuted baselines, MeanPooling and XAttention, respectively, demonstrating the effectiveness of permutation. Concretely, PBS-Attn achieves an average score improvement of 3.21 over MeanPooling on Llama-3.1-8B; this gain is particularly pronounced at longer contexts, reaching an improvement of 7.66 at 128K. PBS-Attn$^+$ further enhances performance, exceeding XAttention by 1.41 on Llama-3.1-8B and 1.05 on Qwen-2.5-7B-1M, approaching the full attention baselines with narrow margins of 3.21 and 0.51, respectively.

**Efficiency Results** To best evaluate the real-world practicality of the sparse attention methods, we measure the end-to-end time to first token (TTFT) on sequence lengths ranging from 8K to 512K. As shown in Figure 3, PBS-Attn achieves the highest speedup across all context lengths, whereas most competing methods only excel within a limited range. For instance, Minference does not show a speedup over FlashAttention until 128k, and the efficiency gains of XAttention stagnate after 128K. Although FlexPrefill matches the speedup of PBS-Attn in most cases, it suffers from a significant quality drop as shown in Table 1 and 2. In contrast, PBS-Attn consistently delivers the best performance, reaching a $2.75\times$ end-to-end speedup at 256K, demonstrating its superior practicality and robustness. To analyze the permutation overhead in PBS-Attn, we further conduct a detailed benchmarking study in Appendix D.

## 4.3 ABLATION STUDIES AND ANALYSIS

**Effect of Permutation** As illustrated in Figure 4, query-aware key permutation consistently increases block-level sparsity by a noticeable margin. For instance, it achieves a 7% absolute sparsity improvement at a context length of 8K, and this gain continues to increase as the context length scales, highlighting the permutation's effectiveness.

**Permutation Target Analysis** To analyze the effect of permutation on queries, we propose a key-aware query permutation approach. However, the attention distribution of queries over keys is often less structured than that of keys over queries. We therefore employ a straightforward strategy that clusters queries which attend to similar keys within a given segment. Specifically, we first compute a set of centroids by calculating block-averaged keys, denoted as $\bar{K}$. Each centroid is defined as $\bar{K}_i = \text{MeanPool}(K_{[(i-1)B+1:iB]})$ for $i = 1, \ldots, T_c$. We then determine cluster assignments by computing the cosine similarity between each query and these centroids. Within each segment,

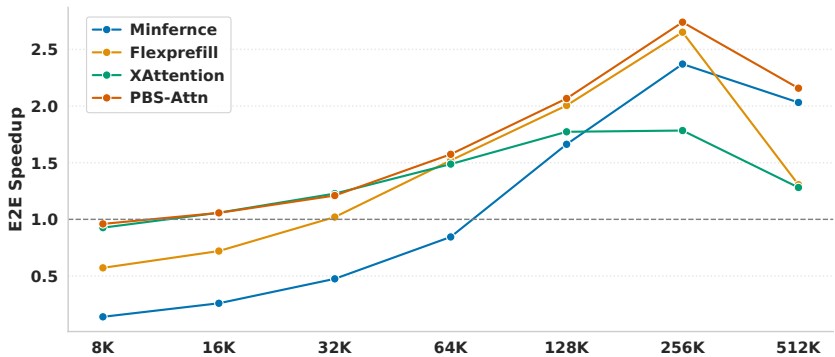

Figure 3: Speedup of various methods relative to FlashAttention, measured by time to first token (TTFT) on LongBenchv2 across various sequence lengths. To accommodate longer sequences under memory constraints, we employ tensor parallelism with tp_size of 2 and 8 for the 256K and 512K contexts, respectively.

queries are assigned greedily based on their similarity to the centroids. We evaluate the effect of the permutation target and order in Figure 5a. The results indicate that permuting both queries and keys brings no noticeable improvements, regardless of the order. Permuting queries offers a marginal improvement over permuting keys in the performance-density trade-off, but it can be less efficient considering the overhead in models with Grouped-Query Attention (GQA) (Ainslie et al., 2023), which have multiple times more query heads than key heads. Accordingly, we exclusively adopt query-aware key permutation in our main method.

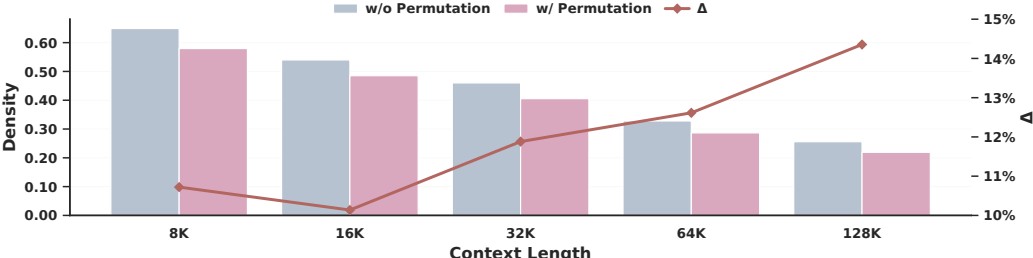

Figure 4: Block-level density on various context lengths with and without permutation. A relative sparsity improvement $\Delta$ is calculated.

**Effect of Segment Size** Segment size $S$ plays a crucial role in segmented permutation, where tokens are permuted within the corresponding segments to maintain inter-segment causality. Intuitively, a larger segment size $S$ takes into account more tokens during sorting, thereby enhancing block-level sparsity; however, it would also include more blocks in the on-diagonal segments, which can not be skipped during computation to avoid breaking causality. Figure 5b illustrates how the segment size, $S$, affects the performance-density trade-off. A larger $S$ flattens the trade-off curve, indicating that segmented permutation effectively clusters key tokens, allowing the model to maintain high performance even at high levels of block-level sparsity. However, this benefit diminishes at lower sparsity levels, as the wide on-diagonal segments contain a large number of blocks that must be computed, limiting block-level sparsity.

**Effect of Block Size** We analyze the impact of block size $B$ on the performance-density trade-off in Figure 5c. Smaller blocks ($B = 64$) provide finer granularity, yielding better performance at very low densities ($< 0.15$) by minimizing redundancy. However, larger blocks ($B = 256$) suffer from rapid degradation at low budgets, as coarse selection forces the inclusion of non-critical tokens. The intermediate size ($B = 128$) strikes the optimal balance, achieving the highest LongBench scores

across most density levels while maintaining robustness. We therefore select $B = 128$ for our main experiments.

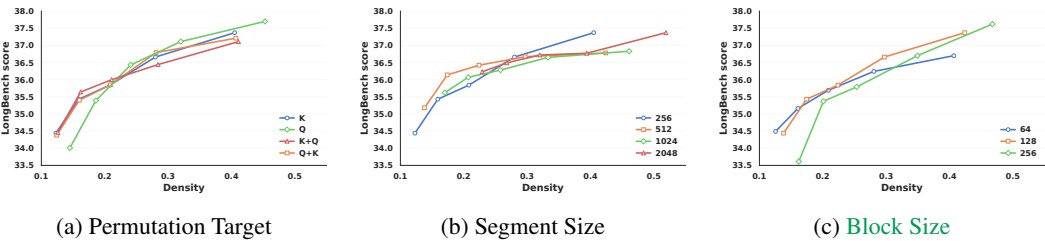

(a) Permutation Target        (b) Segment Size        (c) Block Size

Figure 5: LongBench score vs. average block-level density at a context length of 32K.

## 5 RELATED WORK

**Sparse Attention** The quadratic growth in memory and computational requirements of the attention mechanism has been a bottleneck for scaling LLM context lengths. Sparse attention has emerged as a promising solution, leveraging the inherent sparsity in attention patterns to drastically reduce this overhead. These methods can accelerate different stages of inference, such as prefilling, decoding, or both. StreamingLLM (Xiao et al., 2024) first identifies the attention sink phenomenon in LLMs, proposing to capture a majority of the attention mass with initial and recent tokens. NSA (Yuan et al., 2025) and MoBA (Lu et al., 2025) further incorporate sparse attention into the training stage, accelerating both prefilling and decoding. Methods like H2O (Zhang et al., 2023), can accelerate the decoding speed by exploiting the attention pattern after prefilling. Closely related to this work, various methods are proposed to accelerate the compute-bounded prefilling process. For example, Minference (Jiang et al., 2024) recognizes attention patterns in a pre-computed manner. More recent works tend to perform attention pattern recognition on-the-fly. For instance, FlexPrefill (Lai et al., 2025) utilizes divergence to classify the attention pattern, XAttention (Xu et al., 2025) adopts an antidiagonal scoring metric to weight each block, and SpargeAttention (Zhang et al., 2025) accounts the intra-block similarity into the selection criterion. However, these methods primarily focus on developing better block selection algorithms, while our work is orthogonal: we focus on rearranging the attention matrix to create a structure that inherently increases block-level sparsity.

**Attention with Token Permutation** The idea of reordering tokens to optimize attention computation was pioneered by Reformer (Kitaev et al., 2020), which employs Locality-Sensitive Hashing (LSH) to bucket similar queries and keys, thereby reducing attention computation complexity. However, Reformer relies on a Shared-QK formulation, making it incompatible with modern pre-trained LLMs without significant architectural changes and retraining. In contrast, PBS-Attn is designed as a plug-and-play method and can be applied to any modern LLM without additional training. Concurrent with our work, methods like SVG2 (Yang et al., 2025b) and PAROAttention (Zhao et al., 2025) show promise in accelerating visual generation models like Diffusion Transformers (Peebles & Xie, 2023), but their reliance on bidirectional attention makes them incompatible with the causal constraints of auto-regressive LLMs. PBS-Attn addresses this challenge by introducing a segmented permutation strategy, explicitly preserving inter-segment causality.

## 6 CONCLUSION

In this work, we formalize the permutation properties of the attention mechanism and leverage them to improve block-level sparsity. We introduce Permuted Block-Sparse Attention (PBS-Attn), a plug-and-play method that employs a novel segmented permutation strategy to preserve inter-segment causality while reordering tokens within each segment. Our method achieves an end-to-end prefilling speedup of up to $2.75\times$ with minimal performance degradation, demonstrating a promising path toward more efficient long-context LLMs.

## 7 ETHICS STATEMENT

Our work focuses on improving the computational efficiency of large language models. We believe this research carries positive ethical implications. By reducing the computational resources required for processing long sequences, our method contributes to lowering the energy consumption and carbon footprint associated with training and deploying large-scale AI models. This can also enhance the accessibility of advanced AI technologies, enabling researchers and developers with limited resources to contribute to the field and innovate responsibly.

## 8 REPRODUCIBILITY STATEMENT

To ensure the reproducibility of our work, we commit to making our research as transparent and accessible as possible.

**Code** All code used for our experiments, including the implementation of Permuted Block-Sparse Attention (PBS-Attn) and the custom permuted-FlashAttention kernel, will be made publicly available after the reviewing period. The repository will include scripts to run the evaluations and detailed instructions for setup.

**Models** The experiments were conducted using publicly available state-of-the-art large language models: Llama-3.1-8B (128K) and Qwen-2.5-7B-1M (1M), available for downloading from platforms like HuggingFace.

**Datasets** We used two publicly available and widely recognized benchmarks for long-context language understanding: LongBench and LongBenchv2, available for downloading from platforms like HuggingFace.

**Experimental Setup** All experiments were performed on NVIDIA H100 80GB GPUs. Key implementation details are stated in Section 4.1.

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

# A  PROOFS OF PERMUTATION PROPERTIES

## A.1  PROOF OF LEMMA 3.1

**Lemma A.1** (Key-Value Pair Permutation Invariance). *The attention mechanism is invariant to the order of the source sequence, provided that the key-value pairings are maintained.*

*Formally, let $\boldsymbol{P}_\pi \in \{0,1\}^{M \times M}$ be a permutation matrix that reorders the rows of a matrix according to a permutation $\pi$ on the index set $\{1, \ldots, M\}$. The following identity holds:*

$$Attention(\boldsymbol{Q}, \boldsymbol{P}_\pi \boldsymbol{K}, \boldsymbol{P}_\pi \boldsymbol{V}) = Attention(\boldsymbol{Q}, \boldsymbol{K}, \boldsymbol{V}) \tag{12}$$

*Proof.* Let $\boldsymbol{O} = \text{Attention}(\boldsymbol{Q}, \boldsymbol{K}, \boldsymbol{V})$ and $\boldsymbol{O}' = \text{Attention}(\boldsymbol{Q}, \boldsymbol{P}_\pi \boldsymbol{K}, \boldsymbol{P}_\pi \boldsymbol{V})$. Our goal is to show that $\boldsymbol{O} = \boldsymbol{O}'$. We will prove this by showing that their corresponding row vectors, $\boldsymbol{o}_i$ and $\boldsymbol{o}'_i$, are equal for any arbitrary row index $i \in \{1, \ldots, N\}$.

Let $\boldsymbol{A} = \frac{\boldsymbol{Q}\boldsymbol{K}^T}{\sqrt{d}}$ and $\boldsymbol{W} = \text{softmax}(\boldsymbol{A})$(we use W instead of P as in Eq.1 to avoid confusion) The $i$-th row of the original output is given by:

$$\boldsymbol{o}_i = \sum_{j=1}^{M} \boldsymbol{W}_{ij} \boldsymbol{v}_j$$

Now, let $\boldsymbol{K}' = \boldsymbol{P}_\pi \boldsymbol{K}$ and $\boldsymbol{V}' = \boldsymbol{P}_\pi \boldsymbol{V}$. The score matrix for $\boldsymbol{O}'$ is $\boldsymbol{A}' = \frac{\boldsymbol{Q}(\boldsymbol{K}')^T}{\sqrt{d}} = \frac{\boldsymbol{Q}\boldsymbol{K}^T \boldsymbol{P}_\pi^T}{\sqrt{d}} = \boldsymbol{A}\boldsymbol{P}_\pi^T$. Let $\boldsymbol{W}' = \text{softmax}(\boldsymbol{A}')$.

The $(i, j)$-th element of $\boldsymbol{A}'$ is $\boldsymbol{A}'_{ij} = \sum_{l=1}^{M} \boldsymbol{A}_{il}(\boldsymbol{P}_\pi^T)_{lj} = \boldsymbol{A}_{i,\pi^{-1}(j)}$. The denominator for the softmax computation on the $i$-th row of $\boldsymbol{A}'$ is:

$$\sum_{l=1}^{M} \exp(\boldsymbol{A}'_{il}) = \sum_{l=1}^{M} \exp(\boldsymbol{A}_{i,\pi^{-1}(l)})$$

Since $\pi^{-1}$ is a bijection on $\{1, \ldots, M\}$, this summation is a reordering of the terms $\sum_{k=1}^{M} \exp(A_{ik})$, which is the denominator for the $i$-th row of the original weights $W$.

Thus, the $(i, j)$-th element of the new weight matrix $W'$ is:

$$\boldsymbol{W}'_{ij} = \frac{\exp(\boldsymbol{A}'_{ij})}{\sum_{l=1}^{M} \exp(\boldsymbol{A}'_{il})} = \frac{\exp(\boldsymbol{A}_{i,\pi^{-1}(j)})}{\sum_{k=1}^{M} \exp(\boldsymbol{A}_{ik})} = \boldsymbol{W}_{i,\pi^{-1}(j)}$$

The $i$-th row of the new output $O'$ is a weighted sum of the rows of $V' = P_\pi V$. The $j$-th row of $V'$ is $v'_j = v_{\pi^{-1}(j)}$. Therefore:

$$\boldsymbol{o}'_i = \sum_{j=1}^{M} \boldsymbol{W}'_{ij} \boldsymbol{v}'_j = \sum_{j=1}^{M} \boldsymbol{W}_{i,\pi^{-1}(j)} \boldsymbol{v}_{\pi^{-1}(j)}$$

Let $k = \pi^{-1}(j)$. Since $\pi^{-1}$ is a bijection, summing over all $j \in \{1, \ldots, M\}$ is equivalent to summing over all $k \in \{1, \ldots, M\}$. By this change of variables, we have:

$$\boldsymbol{o}'_i = \sum_{k=1}^{M} \boldsymbol{W}_{ik} \boldsymbol{v}_k = \boldsymbol{o}_i$$

Since $\boldsymbol{o}'_i = \boldsymbol{o}_i$ for an arbitrary $i$, the matrices $\boldsymbol{O}'$ and $\boldsymbol{O}$ are identical. $\square$

## A.2  PROOF OF LEMMA 3.2

**Lemma A.2** (Query Permutation Equivariance). *The attention mechanism is equivariant with respect to permutations of the query sequence.*

*Formally, let $\boldsymbol{P}_\sigma \in \{0,1\}^{N \times N}$ be a permutation matrix that reorders the rows of a matrix according to a permutation $\sigma$ on the index set $\{1, \ldots, N\}$. The following relationship holds:*

$$Attention(\boldsymbol{P}_\sigma \boldsymbol{Q}, \boldsymbol{K}, \boldsymbol{V}) = \boldsymbol{P}_\sigma Attention(\boldsymbol{Q}, \boldsymbol{K}, \boldsymbol{V}) \tag{13}$$

*Proof.* Let $\boldsymbol{O} = \text{Attention}(\boldsymbol{Q}, \boldsymbol{K}, \boldsymbol{V})$ and $\boldsymbol{O}' = \text{Attention}(\boldsymbol{P}_\sigma \boldsymbol{Q}, \boldsymbol{K}, \boldsymbol{V})$. We want to show that $\boldsymbol{O}' = \boldsymbol{P}_\sigma \boldsymbol{O}$.

Let $\boldsymbol{A} = \frac{\boldsymbol{Q}\boldsymbol{K}^T}{\sqrt{d}}$ and $\boldsymbol{W} = \text{softmax}(\boldsymbol{A})$, such that $\boldsymbol{O} = \boldsymbol{W}\boldsymbol{V}$. The score matrix for $\boldsymbol{O}'$ is $\boldsymbol{A}' = \frac{(\boldsymbol{P}_\sigma \boldsymbol{Q})\boldsymbol{K}^T}{\sqrt{d}} = \boldsymbol{P}_\sigma \left( \frac{\boldsymbol{Q}\boldsymbol{K}^T}{\sqrt{d}} \right) = \boldsymbol{P}_\sigma \boldsymbol{A}$. Let $\boldsymbol{W}' = \text{softmax}(\boldsymbol{A}')$.

The softmax function operates independently on each row. Let $(\boldsymbol{X})_i$ denote the $i$-th row of a matrix $\boldsymbol{X}$. Left-multiplication by $\boldsymbol{P}_\sigma$ permutes the rows of $\boldsymbol{A}$, such that the $i$-th row of $\boldsymbol{A}'$ is the $\sigma^{-1}(i)$-th row of $\boldsymbol{A}$: $(\boldsymbol{A}')_i = (\boldsymbol{A})_{\sigma^{-1}(i)}$. Applying the softmax function, the $i$-th row of $\boldsymbol{W}'$ is:

$$(\boldsymbol{W}')_i = \text{softmax}((\boldsymbol{A}')_i) = \text{softmax}((\boldsymbol{A})_{\sigma^{-1}(i)})$$

This resulting vector is identical to the $\sigma^{-1}(i)$-th row of the original weight matrix $\boldsymbol{W}$. Thus, $(\boldsymbol{W}')_i = (\boldsymbol{W})_{\sigma^{-1}(i)}$. This equality for all rows $i$ implies that the entire matrix $\boldsymbol{W}'$ is a row-permuted version of $\boldsymbol{W}$, i.e., $\boldsymbol{W}' = \boldsymbol{P}_\sigma \boldsymbol{W}$.

Now we can write the output $\boldsymbol{O}'$ as:

$$\boldsymbol{O}' = \boldsymbol{W}'\boldsymbol{V} = (\boldsymbol{P}_\sigma \boldsymbol{W})\boldsymbol{V}$$

By the associativity of matrix multiplication, we have:

$$\boldsymbol{O}' = \boldsymbol{P}_\sigma (\boldsymbol{W}\boldsymbol{V}) = \boldsymbol{P}_\sigma \boldsymbol{O}$$

This completes the proof. $\qquad\square$

### A.3 PROOF OF THEOREM 3.3

**Theorem A.3** (Attention Permutation Invariance under Inverse Transformation). *If the queries are permuted by $\boldsymbol{P}_\sigma$ and the key-value pairs are permuted by $\boldsymbol{P}_\pi$, the resulting output is a permuted version of the original output. Applying the inverse of the query permutation recovers the original, unpermuted output. Formally:*

$$\boldsymbol{P}_\sigma^T \text{ Attention}(\boldsymbol{P}_\sigma \boldsymbol{Q}, \boldsymbol{P}_\pi \boldsymbol{K}, \boldsymbol{P}_\pi \boldsymbol{V}) = \text{Attention}(\boldsymbol{Q}, \boldsymbol{K}, \boldsymbol{V}) \tag{14}$$

*Proof.* We prove the theorem by showing that the left-hand side (LHS) of the equation simplifies to the right-hand side (RHS) through sequential application of the preceding lemmas.

$$
\begin{aligned}
\text{LHS} &= \boldsymbol{P}_\sigma^T \text{ Attention}(P_\sigma Q, P_\pi K, P_\pi V) && \\
&= \boldsymbol{P}_\sigma^T \text{ Attention}(\boldsymbol{P}_\sigma Q, K, V) && \text{by Lemma 3.1} \\
&= \boldsymbol{P}_\sigma^T (\boldsymbol{P}_\sigma \text{ Attention}(Q, K, V)) && \text{by Lemma 3.2} \\
&= (\boldsymbol{P}_\sigma^T \boldsymbol{P}_\sigma) \text{ Attention}(Q, K, V) && \text{by associativity} \\
&= I \cdot \text{Attention}(Q, K, V) && \text{since } P_\sigma \text{ is orthogonal} \\
&= \text{Attention}(Q, K, V) && \\
&= \text{RHS} &&
\end{aligned}
$$

The final expression is identical to the right-hand side, which concludes the proof. $\qquad\square$

## B   PERMUTATION EFFECT ON BLOCK-LEVEL SPARSITY

In this section, we analyze how key token permutation can be leveraged to improve block-level sparsity. As discussed in the main text, attention is inherently sparse: for a given query token $q_i$, a small subset of key tokens $\mathcal{S}_i \subset \{k_1, \ldots, k_N\}$ accounts for the majority of the attention mass. However, block-sparse attention operates at a coarser granularity. If the tokens in $\mathcal{S}_i$ are scattered throughout the sequence, the mechanism must retrieve every block containing keys in $k \in \mathcal{S}_i$, leading to redundancy in block computation and inefficiency. Consequently, for a specific block of queries indexed by $B$, the set of all critical key tokens $\mathcal{K}_B$ is the union of the individual subsets:

$$\mathcal{K}_B = \bigcup_{i \in B} \mathcal{S}_i \tag{15}$$

Given a sparsity level $\alpha$, the objective of block-sparse attention is to maximize the total attention mass captured by the selected key blocks $\mathbb{C}_{\text{sel}}$:

$$\max_{\mathbb{C}_{\text{sel}}} \sum_{C \in \mathbb{C}_{\text{sel}}} \sum_{i \in B} \sum_{j \in C} A_{i,j} \quad \text{s.t.} \quad |\mathbb{C}_{\text{sel}}| \leq m_\alpha \tag{16}$$

where $A_{i,j}$ is the oracle token-level attention matrix and $m_\alpha$ is the maximum budget of key blocks to be selected at sparsity level $\alpha$.

Intuitively, since permutation serves as a coarse-grained token selection process, it should be beneficial to leverage query information when permuting keys, and vice versa. Here, we compare five different permutation strategies and calculate the attention coverage using Eq. 16:

- **No Permutation**: serves as the baseline, where queries and keys keep their original ordering.

- **Random Permutation**: randomly shuffles tokens within each segment to show that permutation alone does not improve coverage without guidance.

- **Greedy Query-aware Key Permutation**: within each segment, compute block centroids of queries via mean pooling with block size $B$, then iteratively assign the $B$ most similar key tokens (cosine similarity) to each centroid. Note that we also ablated the effect of permuting queries using a symmetric version of this strategy in Section 4.3.

- **Random-Query-Attention-based Key Permutation**: sample $B$ queries at random, calculate and average their attention scores to the keys, and rank the keys within each segment according to the resulting attention scores.

- **Last-Block-Query-Attention-based Key Permutation**: our default strategy that uses the last block of queries to compute attention scores and reorder keys according to the averaged attention scores within each segment.

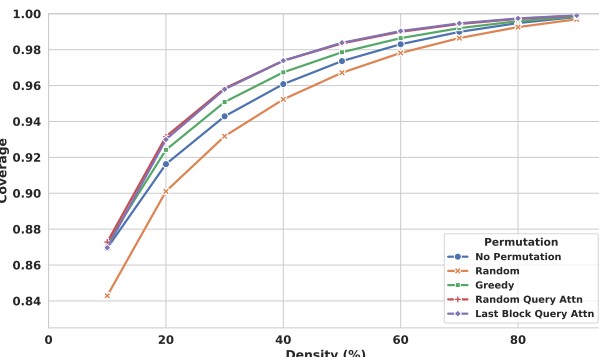

Figure 6: Coverage-density trade-off of various permutation methods. The results are measured with Llama-3.1-8B with a context length of 16K.

**Comparision of Permutation Methods**    Figure 6 illustrates the attention coverage across various block-level density budgets. The results indicate the following:

- **The coverage-density distribution exhibits a long-tailed pattern**, indicating that achieving higher coverage necessitates the retrieval of a significantly larger number of key blocks.

- Random Permutation degrades attention coverage compared to the baseline. **Permutation without heuristics disrupts the natural local token similarities of LLMs, leading to lower block-level sparsity.**

- Greedy Query-aware Key Permutation improves attention coverage over the baseline, **demonstrating that leveraging local query information to permute keys helps concentrate the attention mass**, thereby achieving higher block-level sparsity.

- Both "Query-Attention-based" strategies achieve the highest coverage, outperforming both the greedy approach and the baseline. This suggests that sorting keys by their accumulated attention scores effectively identifies and groups the "heavy hitter" keys(defined in Zhang et al. (2023), also

recognized as "Vertical Lines" in previous literature (Jiang et al., 2024; Lai et al., 2025)). Furthermore, this proves more effective than utilizing local query centroids (as in the greedy approach), indicating that the **permutation benefits most from grouping key tokens that are globally critical to all queries.**

- The performance of Random-Query-Attention and Last-Block-Query-Attention is nearly identical. This implies that **the identification of critical key tokens is robust to the specific subset of queries used for scoring**. Consequently, using the last block of queries is validated as an efficient, practical proxy for global key importance.

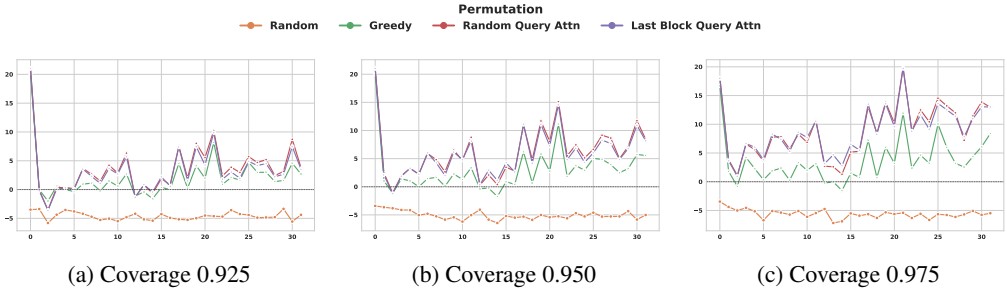

(a) Coverage 0.925        (b) Coverage 0.950        (c) Coverage 0.975

Figure 7: Layer-wise absolute sparsity improvement at various coverage levels. This metric calculates the sparsity improved by permutation. For example, if the baseline requires 60% block density to achieve 0.95 coverage, while the permuted method requires only 40%, the recorded sparsity improvement is 20%. Results are measured with Llama-3.1-8B with a context length of 16K.

**Layer-wise Sparsity Improvement** Figure 7 illustrates the layer-wise absolute sparsity improvement at various coverage levels. For the permutation method comparisons, the results align with the findings in Figure 6 across all layers. Strategies leveraging query attention consistently improve sparsity compared to Random Permutation and the baseline; moreover, this improvement becomes more significant at higher coverage levels. This confirms that the proposed permutation strategies effectively group critical key tokens, which is particularly beneficial given the long-tailed nature of the coverage-density distribution (Figure 6). In the layer-wise breakdown, **Layer 0 consistently exhibits high sparsity improvement**. This indicates that the "heavy hitter" phenomenon (or the "Vertical Lines" in the attention map) is especially prominent in the first layer, where permutation successfully consolidates these globally attended keys. **For the remaining layers, the sparsity improvement scales with the coverage level.** This suggests that for most of the layers (especially the middle-to-deep layers), the primary benefit of permutation stems from clustering the scattered "heavy hitter" tokens located in the tail of the attention mass distribution, which are otherwise expensive to retrieve.

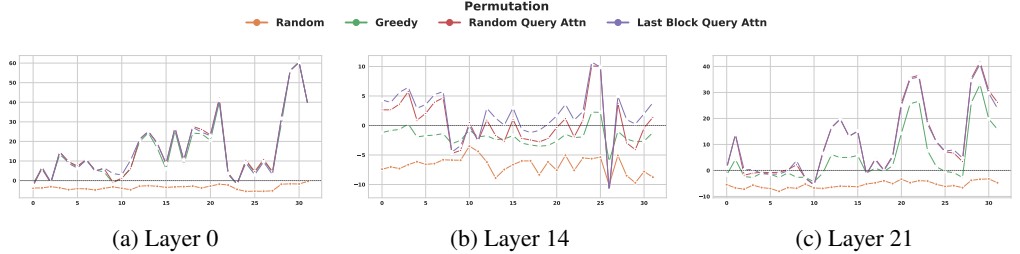

(a) Layer 0        (b) Layer 14        (c) Layer 21

Figure 8: Head-wise absolute sparsity improvement of representative layers at attention coverage of 0.975.

**Head-wise Sparsity Improvement** Figure 8 shows the absolute head-wise sparsity improvement for three representative layers with an attention coverage of 0.975. The results reveal diverse responses to permutation across layers and heads. In the first layer, which consistently benefits from permutation, nearly all heads become sparser (Figure 8a), with some showing substantial gains (e.g.,

Head 30 shows a 60% absolute improvement). In most other layers, the vast majority of heads improve with permutation. However, we identified a few outliers; for example, Head 26 in Layer 14 (Figure 8b) becomes denser, resulting in only a marginal overall improvement for that layer. In contrast, other layers like Layer 21 (Figure 8c) lack these negatively affected heads, and their mix of insensitive and improved heads leads to a noticeable overall increase in sparsity.

**Failure Mode Analysis**   Here we analyze why certain attention heads exhibit marginal improvement or even degraded sparsity under permutation by visualizing their attention maps. Zooming in on Figure 8, we select two representative cases: Layer 14 Head 26 (negative sparsity gain) and Layer 21 Head 2 (marginal gain). For the minority of heads dominated by the "Slash Line" pattern (Figure 9a), a phenomenon also recognized in previous literature Jiang et al. (2024); Lai et al. (2025) where queries attend to keys at fixed intervals, permutation fails to improve sparsity. This occurs because selecting the corresponding diagonal blocks is naturally the optimal strategy to cover "Slash Lines". Any permutation inevitably disrupts this structure, scattering the keys and leading to redundancy in block selection. Regarding heads showing highly query-specific patterns where different queries attend to distinct sets of keys (Figure 9b), the sparsity improvement from permutation remains marginal. In contrast, permutation yields significant improvements for the majority of heads where most queries attend to the same shared set of keys (Figure 2). Consequently, the overall sparsity improvement could be further enhanced by incorporating pruning strategies to exclude the few heads with negative sparsity gains, which we leave for future work.

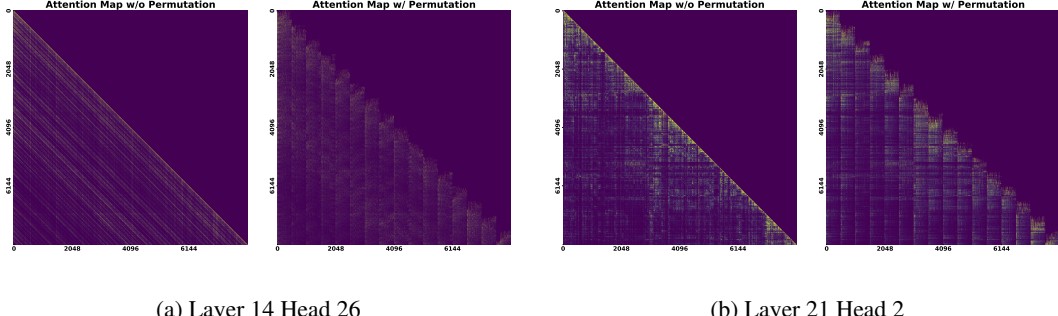

(a) Layer 14 Head 26                         (b) Layer 21 Head 2

Figure 9: Visualization of attention maps for heads without noticeable sparsity gains.

## C   BLOCK SELECTION

### C.1   BLOCK SELECTION IN PBS-ATTN

We use a mean pooling strategy and block-wise attention to estimate the importance of each key block. This method is also used for unpermuted sequences, serving as a strong baseline denoted as MeanPooling in the main paper. Here we detail the implementation of MeanPooling selection in Algorithm 2. Note that for the baseline MeanPooling, $Q'$ and $K'$ remain unpermuted as $Q' = Q$ and $K' = K$. The causal mask $C$ is a upper triangular matrix with entries set to $-\infty$. If segmented permutation is applied, this mask also includes the on-diagonal segments (as in Figure 1), to ensure valid intra-segment attention post-permutation.

---

**Algorithm 2** MeanPooling Block Selection

---

**Require:** Query matrix $Q' \in \mathbb{R}^{N \times d}$, Key matrix $K' \in \mathbb{R}^{N \times d}$, block size $B$, attention score threshold $\tau$, causal mask $C \in \{0, -\infty\}^{\lceil N/B \rceil \times \lceil N/B \rceil}$.
**Ensure:** Block selection mask $M \in \{0, 1\}^{\lceil N/B \rceil \times \lceil N/B \rceil}$.
  1: Divide $Q'$, $K'$ into blocks of size $B$: $\{Q'_i\}_{i=1}^{T_r}$, $\{K'_j\}_{j=1}^{T_c}$, where $T_r = T_c = \lceil N/B \rceil$.
  2: Compute pooled queries: $\bar{Q}_i = \text{MeanPool}(Q'_i)$ for $i = 1, \ldots, T_r$.
  3: Compute pooled keys: $\bar{K}_j = \text{MeanPool}(K'_j)$ for $j = 1, \ldots, T_c$.
  4: Form pooled matrices $\bar{Q} \in \mathbb{R}^{T_r \times d}$ and $\bar{K} \in \mathbb{R}^{T_c \times d}$.
  5: Compute block scores: $S_{\text{block}} = \text{softmax}(\bar{Q}\bar{K}^T/\sqrt{d} + C)$.
  6: Initialize $M = 0$.
  7: **for** $i = 1$ to $T_r$ **do**
  8:     Get scores for query block $i$: $a_i = S_{\text{block}}[i, 1:i]$.
  9:     Sort scores and get original indices: $o_i = \text{argsort}(-a_i)$.
10:     Compute cumulative sum on sorted scores: $c_i = \text{cumsum}(a_i[o_i])$.
11:     Find number of blocks to select: $k = \min(\{j \mid c_i[j] \geq \tau\} \cup \{i\})$.
12:     Get indices of blocks to select: $\mathcal{J} = o_i[1:k]$.
13:     Set $M[i, j] = 1$ for all $j \in \mathcal{J}$.
14: **end for**
15: **return** $M$.

---

### C.2 PBS-ATTN WITH EXISTING BLOCK SELECTION ALGORITHMS

In the main paper, we use a simple mean pooling strategy for block selection in block-sparse attention, as detailed in Section C.1, and show that permutation can increase block-level sparsity under this naive mean pooling strategy (Section 4.3). In this section, we further demonstrate that advanced block selection algorithms (e.g. XAttention) can also benefit from permutation.

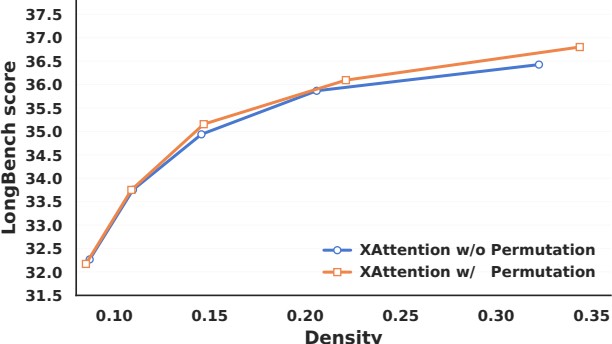

Figure 10: Longbench score vs. average block-level density at a context length of 32k of XAttention selection with and without permutation.

As shown in Figure 10, XAttention selection can also benefit from the sparsity improvements of permutation, achieving a better trade-off between performance and sparsity.

## D ANALYSIS ON THE PERMUTATION OVERHEAD

**Time Overhead**    As shown in Figures 11a and 11b, the permutation overhead in PBS-Attn is negligible compared to the main attention computation time, especially at longer context lengths. For instance, at a context length of 128K, permutation introduces an overhead of only $4\%$ relative to the block attention computation time and just $1.3\%$ compared to FlashAttention. While permuting queries introduces a slightly higher overhead than permuting keys, this difference diminishes as the context length increases. However, query permutation can also result in lower block-level sparsity than key permutation under the same settings, leading to higher attention computation time. Detailed

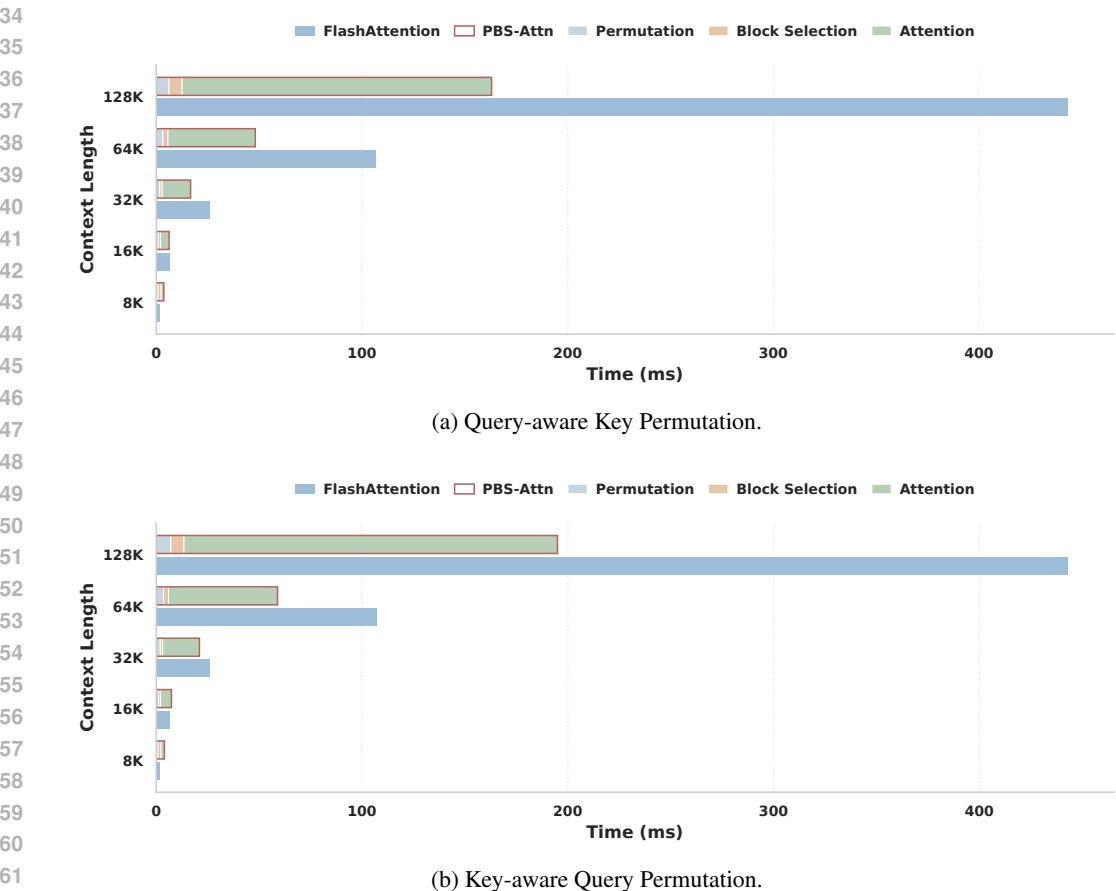

(a) Query-aware Key Permutation.

(b) Key-aware Query Permutation.

Figure 11: Detailed benchmarking results of PBS-Attn vs. FlashAttention.

benchmarking results are shown in Table 4. At a 512K context length, the permutation overhead (i.e., permutation time plus block selection time) is only $3.1\%$, while providing a $3.41\times$ attention speedup, further demonstrating the practical potential of PBS-Attn.

**Memory Overhead**  Here we analyze the memory overhead of PBS-Attn. For the proposed Last-Block-Query Key Permutation strategy, we require only the last block of queries to calculate proxy scores; consequently, the memory cost for scoring scales linearly with context length. Specifically, given block size $B$ and context length $N$, the memory cost for scoring is $O(B \times N)$. For $B = 128$ and $N = 128$K, this amounts to approximately 32MiB per head in BFloat16, which is negligible relative to the total activation memory for long sequences. Regarding the memory overhead during permutation, our current implementation explicitly creates physically permuted key and value tensors using a `torch.gather()` call, which allocates a temporary buffer proportional to the sequence size. For Llama-3.1-8B with head dimension $d = 128$, this results in an additional 32MiB per head in BFloat16, which is also insignificant. Furthermore, this overhead could be mitigated via index remapping, allowing the attention kernel to retrieve data directly from the original vectors using permuted indices. Nonetheless, since the prefilling phase of LLMs is primarily compute-bound, the impact of this memory movement is minimal. As shown in Table 4, the relative overhead decreases further in memory-intensive longer context scenarios. Figure 13 further demonstrates that PBS-Attn maintains consistent speedups on larger models (e.g., Qwen-2.5-14B) despite their higher memory requirements, confirming that memory overhead does not become a bottleneck as model size scales.

# E  GQA HANDLING

Table 4: Timing breakdown (ms) for PBS-Attn relative to FlashAttention. Measured by profiling on CUDA events on an H100 80GB GPU.

| Length | FlashAttention | Permutation | Block Selection | Attention | Total | Overhead | Speedup |
|--------|----------------|-------------|-----------------|-----------|-------|----------|---------|
| _Key Permutation_ | | | | | | | |
| 4K | 0.54 | 0.72 | 0.84 | 0.53 | 2.09 | 74.6% | 0.26× |
| 8K | 1.78 | 0.82 | 0.86 | 1.31 | 2.99 | 56.2% | 0.60× |
| 16K | 6.67 | 1.02 | 0.62 | 4.08 | 5.71 | 28.7% | 1.17× |
| 32K | 26.10 | 1.68 | 0.60 | 13.83 | 16.11 | 14.2% | 1.62× |
| 64K | 106.84 | 3.24 | 1.24 | 42.44 | 46.92 | 9.5% | 2.28× |
| 128K | 443.29 | 6.22 | 3.21 | 150.45 | 159.87 | 5.9% | 2.77× |
| 256K | 1837.32 | 13.87 | 11.81 | 563.54 | 589.22 | 4.4% | 3.12× |
| 512K | 7496.67 | 26.76 | 41.99 | 2128.16 | 2196.91 | 3.1% | 3.41× |
| _Query Permutation_ | | | | | | | |
| 4K | 0.54 | 0.70 | 0.81 | 0.63 | 2.31 | 65.4% | 0.23× |
| 8K | 1.78 | 0.82 | 0.81 | 1.67 | 3.30 | 49.4% | 0.54× |
| 16K | 6.67 | 1.20 | 0.49 | 5.06 | 6.75 | 25.0% | 0.99× |
| 32K | 26.10 | 1.98 | 0.59 | 17.81 | 20.38 | 12.6% | 1.28× |
| 64K | 106.84 | 3.52 | 1.25 | 52.95 | 57.72 | 8.3% | 1.85× |
| 128K | 443.29 | 7.10 | 3.23 | 181.51 | 191.85 | 5.4% | 2.31× |
| 256K | 1837.32 | 13.82 | 11.80 | 597.04 | 622.66 | 4.1% | 2.95× |
| 512K | 7496.67 | 26.83 | 41.91 | 2481.68 | 2550.42 | 2.7% | 2.94× |

Modern LLMs often employ Grouped Query Attention (GQA), where a group of query heads shares the same key and value heads to reduce inference overhead. In this section, we compare two different GQA handling strategies for PBS-Attn: (1) the **default strategy**, where we replicate the keys and values for each query head in a group to apply unique permutations, and (2) the **shared permutation strategy**, where we average the queries across the head dimension within each group to compute a single permutation, ensuring that all queries within the same group share the keys and values in the same order.

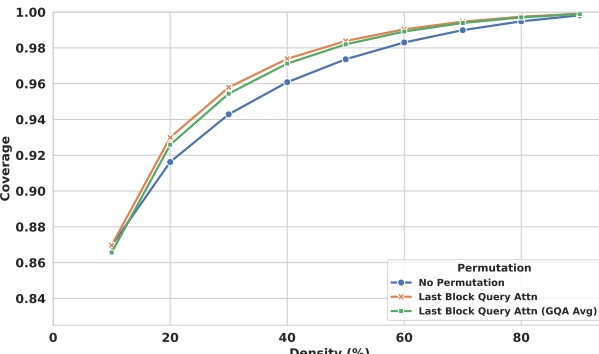

Figure 12: Coverage-density trade-off with two different GQA handling strategies. The results are measured with Llama-3.1-8B with a context length of 16K.

Figure 12 illustrates the trade-off between attention coverage and density for these two GQA handling strategies. The results imply that sharing the permutation within a GQA group affects sparsity only marginally, while still maintaining a significant coverage gain compared to the no-permutation baseline. We further evaluate this approach on real-world datasets using LongBench. As shown in Table 5, the shared permutation strategy achieves performance comparable to the default strategy, closely aligning with the findings in Figure 12. This demonstrates that sharing the permutation within a GQA group has minimal impact on sparsity gains and practical performance, suggesting an more efficient approach for the deployment of PBS-Attn.

# F    EVALUATION ON QWEN-2.5-14B-1M

Table 5: Performance comparison with different GQA handling strategies on LongBench.

| Method | Single-Doc QA | Multi-Doc QA | Summarization | Few-shot Learning | Code | Synthetic | Avg. |
|---|---|---|---|---|---|---|---|
| **PBS-Attn(Default)** | 48.00 | 42.09 | 17.72 | 28.36 | 24.25 | 63.80 | 37.37 |
| **PBS-Attn(Shared Permutation)** | 48.38 | 41.47 | 17.84 | 27.77 | 23.86 | 63.92 | 37.21 |

To further validate the effectiveness of PBS-Attn on larger LLMs, we conduct evaluations using Qwen-2.5-14B-1M on the LongBench benchmark. As presented in Table 6, PBS-Attn consistently outperforms the baselines; this is consistent with the results observed in Table 1 and confirms the scalability of PBS-Attn to larger LLMs. Regarding efficiency, Figure 13 illustrates that PBS-Attn achieves nearly a $2\times$ speedup at a context length of 128K. This trajectory closely matches that of the 7B model, further demonstrating the method's efficiency at scale. Note that the speedup at the 256K context length cannot be directly compared to the 7B model results due to differing tensor parallelism settings required by memory constraints.

Table 6: Performance comparison of various sparse attention methods on LongBench with Qwen-2.5-14B-1M. **Bold** and underlined scores indicate the best and second-best performing methods in each category, respectively, with the exception of the full attention baseline.

| Method | Single-Doc QA | Multi-Doc QA | Summarization | Few-shot Learning | Code | Synthetic | Avg. |
|---|---|---|---|---|---|---|---|
| Full | 47.33 | 47.44 | 15.55 | 59.13 | 18.67 | 71.33 | 43.24 |
| MInference | 46.50 | 45.73 | 15.56 | 57.23 | 18.76 | 63.33 | 41.19 |
| FlexPrefill | 44.73 | 43.37 | 15.62 | 53.55 | 9.88 | 35.00 | 33.69 |
| XAttention | **46.65** | 46.03 | **15.63** | **58.69** | **19.56** | 63.33 | 41.65 |
| MeanPooling | 45.41 | 45.57 | 15.58 | 57.40 | 17.25 | 35.56 | 36.13 |
| **PBS-Attn** | 46.56 | **46.43** | 15.48 | 58.51 | 17.53 | **67.17** | **41.95** |

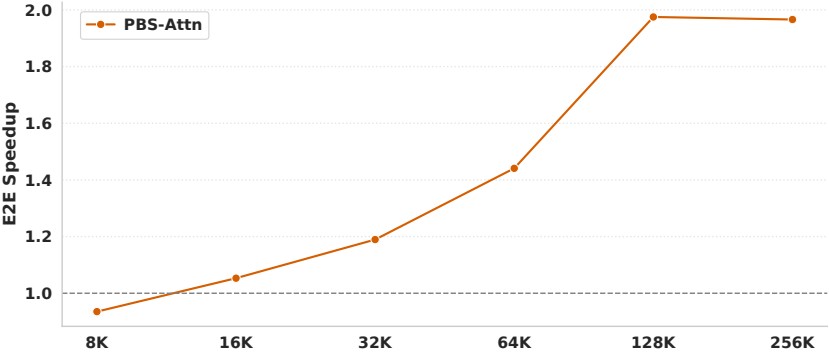

Figure 13: Speedup of PBS-Attn relative to FlashAttention on various context lengths. We employ tensor parallelism with tp_size of 4 for 256K context due to memory constraints.

## G  VISUALIZATION OF PERMUTATION

In this section, we provide more visualizations of the permutation effect on both Llama-3.1-8B (Figure 14) and Qwen-2.5-7B-1M (Figure 15).

## H  USE OF LARGE LANGUAGE MODELS

During the preparation of this work, we utilized large language models (LLMs) to assist with code development and manuscript writing. Specifically, their applications included improving the grammar and clarity of the text, as well as assisting in code completion.

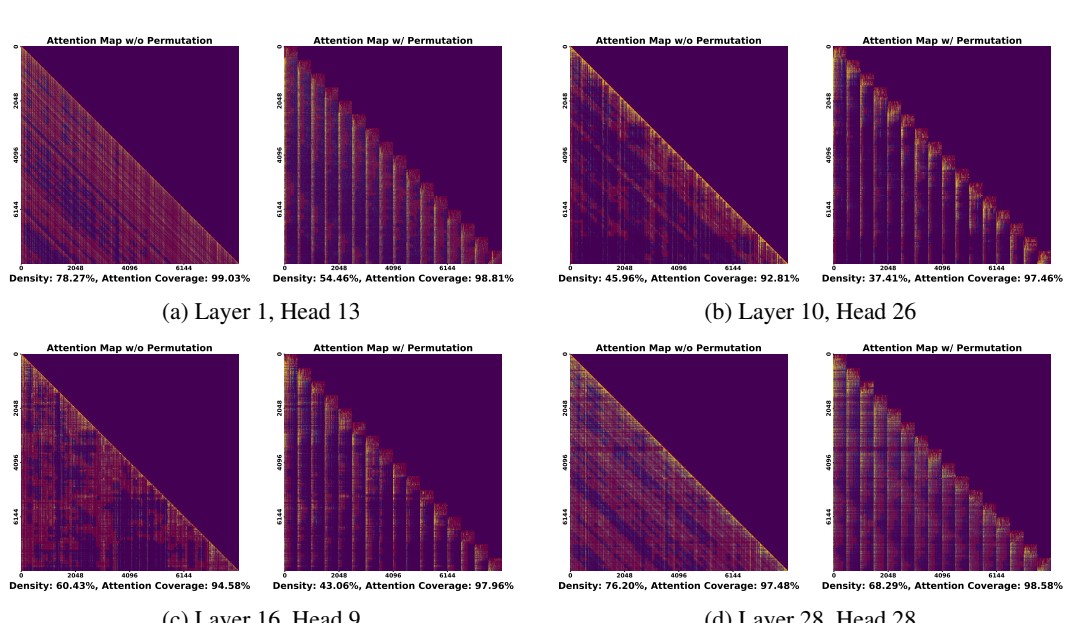

Figure 14: Permutation visualizations of Llama-3.1-8B.

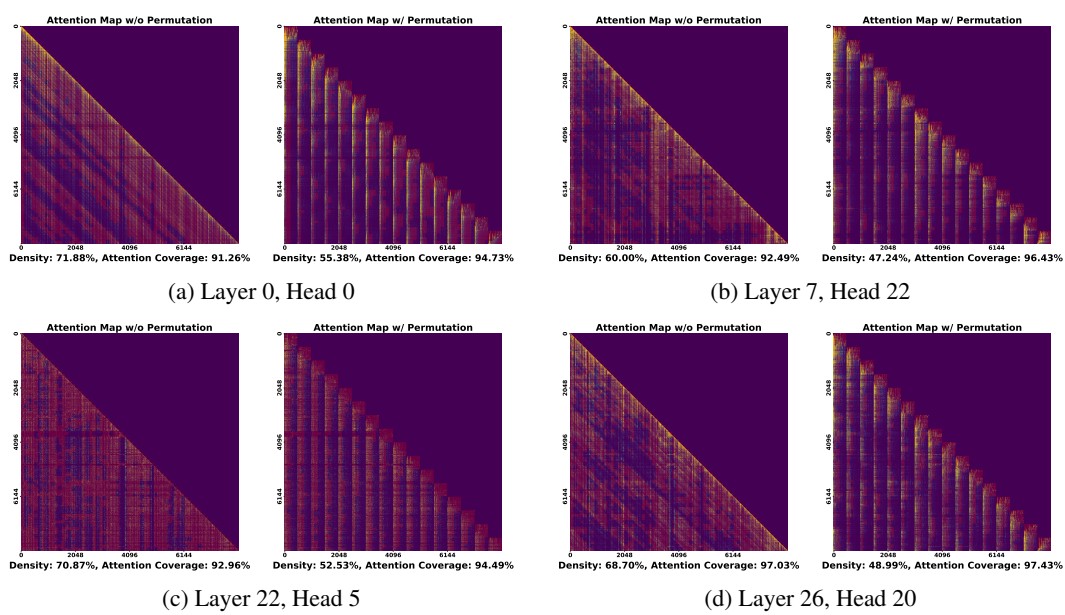

Figure 15: Permutation visualizations of Qwen-2.5-7B-1M.

