# OpenReview forum: "Sparser Block-Sparse Attention via Token Permutation"
_ICLR.cc/2026/Conference — Submitted to ICLR 2026_

### Official Review · Reviewer_FuSK · 2025-10-18

**Soundness:** 1
**Presentation:** 2
**Contribution:** 1
**Rating:** 2
**Confidence:** 4

**Summary:**

The paper proposes Permuted Block-Sparse Attention (PBS-Attn), a method leveraging permutation invariance in the attention mechanism to rearrange query and key tokens, thereby improving block-level sparsity and efficiency for long-context large language models (LLMs). The authors claim that, by performing segmented permutations that preserve causality, the method achieves up to 2.75× speedup over FlashAttention with little loss in accuracy on LongBench and LongBenchv2.

**Strengths:**

1. Provides a formally correct analysis of permutation invariance in the attention mechanism.
2. Introduces an efficient implementation using Triton kernels, showing measurable runtime gains.

**Weaknesses:**

1. The paper does not adequately explain why permutation improves block-wise sparsity.
Prior works (e.g., StreamingLLM, Minference, FlexPrefill, SampleAttention) discuss distinct sparsity patterns in long-context attention—recent-token, column (“vertical”), or slash patterns.
PBS-Attn only demonstrates improved sparsity empirically without analyzing how permutation aligns with these known patterns. Without this, the observed gains appear heuristic rather than principled.
2. In sparse attention, which tokens are retained is crucial for model accuracy.
The proposed “query-aware key permutation” seems to rely on average attention from the last query block, but the rationale, robustness, and comparison with established token-importance metrics are missing.
Consequently, it remains unclear why the method preserves accuracy relative to full attention.
3. The paper reports that PBS-Attn not only speeds up computation but also improves accuracy compared to other sparse methods.
However, the source of this improvement is not analyzed.
3. The description of how permutations are implemented in memory is vague.
If permutation involves physically reordering the K/V cache or attention tensors, the memory movement could dominate runtime, negating theoretical speedups.
The paper should explicitly clarify whether permutations are handled via index mapping (logical reordering) or physical memory rearrangement. As attention is typically memory-bound, even small data shuffling can be expensive.
4. Despite the formal theorem, the proposed method largely appears as an engineering improvement to block-sparse attention rather than a conceptual advance in understanding attention sparsity.
There is little theoretical or empirical insight into how permutation interacts with attention distribution, context structure, or causal masking beyond its computational benefit.

**Questions:**

See weaknesses.

---

> ### Author Response · Authors · 2025-11-27
>
> We thank you for your critical and stimulating feedback. We will address your concerns below.
>
> > **W1:** The paper does not adequately explain why permutation improves block-wise sparsity. Prior works (e.g., StreamingLLM, Minference, FlexPrefill, SampleAttention) discuss distinct sparsity patterns in long-context attention—recent-token, column (“vertical”), or slash patterns. PBS-Attn only demonstrates improved sparsity empirically without analyzing how permutation aligns with these known patterns. Without this, the observed gains appear heuristic rather than principled.
> >
>
> **A1:**
>
> Thank you for this insightful comment. We agree that a principled explanation of *how* permutation interacts with established attention patterns is essential. **In short, the sparsity gain in PBS-Attn primarily stems from clustering globally important key tokens** (also known as “heavy hitter” tokens in H2O [1] or “Vertical Lines” in Minference [2] and FlexPrefill [3]).
>
> To better illustrate this, **we have significantly expanded Section 3.3 and added a comprehensive analysis of the permutation effect on block sparsity in Appendix B of the revision.**
>
> In this added section, we first formalize block-level coverage in block-sparse attention (Eq. 16). Based on this, we perform a statistical evaluation using Llama-3.1-8B with 16K inputs (the maximum feasible length for materializing the full oracle attention matrix within our memory budget) across three axes: method-wise, layer-wise, and head-wise.
>
> The overall benefit is mainly illustrated by the method-wise comparison. We introduce four baselines:
>
> 1. No Permutation (acts as the unpermuted baseline).
> 2. Random Permutation (to observe how permutation without guidance affects sparsity).
> 3. Greedy-Query-aware Permutation  (which leverages local query information by computing centroids and performing a strict key token assignment).
> 4. Random-Query-Attn Permutation (which selects a random set of queries to score the keys).
>
> **We compare the proposed Last-Block-Query-Attn with these four baselines and plot their coverage-density (sparsity) trade-off in Figure 6. At the same sparsity level, the attention coverage follows this ranking:   Last-Query-Attn  $\approx$ Random-Query-Attn > Greedy > No Permutation > Random Permutation**
>
> This demonstrates that:
>
> 1. Permutation without guidance disrupts local token similarities, leading to degradation (No Permutation > Random Permutation).
> 2. Using query information to permute keys improves coverage at certain sparsity levels (Last-Query-Attn/Random-Query-Attn/Greedy > No Permutation).
> 3. Permuting Keys via a global sorting leads to better sparsity than grouping tokens via local query information (Last-Query-Attn/Random-Query-Attn > Greedy). **This confirms that it best benefits the sparsity via clustering the globally important keys rather than a specific query-aware scheme, which is also shown in Figure 2 in the paper.**
> 4. The global sorting is robust to the selection of the query subset, since we’re finding keys that are critical to all queries (Last-Query-Attn  $\approx$ Random-Query-Attn).
>
> In the layer-wise comparisons in Section B (Figure 7), where we directly compare the sparsity gain at different coverage levels, **there is a clear trend for most layers (except Layer 0) where sparsity gain increases with the coverage level.** This occurs because the attention distribution is inherently long-tailed (as seen in Figure 6); **permutation successfully groups the scattered “heavy hitter” tokens, thereby reducing the number of blocks that need to be retrieved.**
>
> Finally, we incorporate head-wise comparisons and a failure mode analysis. As shown in Figure 8, most heads exhibit a noticeable gain via permutation, with the exception of some outliers. We further analyze these outliers by visualizing their attention maps (Figure 9). We observe that for highly query-specific heads (where queries tend to attend to different sets of keys), the sparsity gain from permutation is marginal. For heads dominated by the “Slash Line” attention pattern (where queries attend to keys at fixed intervals), permutation can lead to sparsity degradation. This is expected, as selecting the corresponding diagonal blocks is naturally the optimal strategy to cover slash lines, and any permutation will disrupt this structure. Nonetheless, these outliers account for only a minority of heads (Figure 8), and the overall sparsity increases. Performance could be further enhanced by incorporating pruning strategies that leave these outlier heads unpermuted.
>
> For more details, please refer to Section 3.3 and Appendix B in the revised paper. Thank you again for your effort to improve our work!
>
> [1] H2O: Heavy-Hitter Oracle for Efficient Generative Inference of Large Language Models
>
> [2] MInference 1.0: Accelerating Pre-filling for Long-Context LLMs via Dynamic Sparse Attention
>
> [3] FlexPrefill: A Context-Aware Sparse Attention Mechanism for Efficient Long-Sequence Inference

---

> > ### Author Response · Authors · 2025-11-27
> >
> > > **W2:** In sparse attention, which tokens are retained is crucial for model accuracy. The proposed “query-aware key permutation” seems to rely on average attention from the last query block, but the rationale, robustness, and comparison with established token-importance metrics are missing. Consequently, it remains unclear why the method preserves accuracy relative to full attention.
> > >
> >
> > **A2:**
> >
> > Thank you for this critical question.
> >
> > As discussed in **A1** and **Appendix B**, the rationale behind the proposed “Last-Block-Query-aware Key Permutation” is to increase block sparsity by clustering globally important key tokens. This approach has been shown to be robust to the selection of queries; specifically, our results demonstrate that a random set of queries can effectively serve as the scoring proxy (**Figure 6**), making the use of the last block of queries a valid and practical choice.
> >
> > Regarding the comparison with established token-importance metrics, we would like to clarify that **our work operates within the scope of block-sparse attention**, similar to FlexPrefill [3] and XAttention [4]. These established works focus on *how to weigh and select* blocks (e.g., using block-level attention scores or antidiagonal scoring) to cover all the critical tokens.  **Our paper takes a different perspective: we group critical keys into fewer blocks, thereby reducing the redundancy of block selection. This approach is complementary to existing selection methods**, as we showed that permutation can be combined with algorithms like XAttention to achieve even higher sparsity (**Section C.2**).
> >
> > **Furthermore, by grouping key tokens with similar behaviors, our proposed permutation makes blocks more distinct from one another, facilitating more effective selection.** Concretely, for Qwen-2.5-7B-1M, PBS-Attn achieves a significant performance improvement of **+4.54 on LongBench (Table 1)** and **+8.15 on LongBenchv2 (Table 2)**, despite sharing the same block selection algorithm (MeanPooling + block-level attention) as the baseline. This further demonstrates the effectiveness of our permutation method. PBS-Attn maintains the performance of full attention primarily by achieving higher attention coverage at equivalent sparsity levels via permutation. When coverage is sufficiently high, the model is able to preserve its original performance.
> >
> > [4] XAttention: Block Sparse Attention with Antidiagonal Scoring
> >
> > > **W3:** The paper reports that PBS-Attn not only speeds up computation but also improves accuracy compared to other sparse methods. However, the source of this improvement is not analyzed.
> > >
> >
> > **A3:**
> >
> > Thank you for this suggestion. **The source of the accuracy improvement is the structural optimization of the attention matrix provided by permutation.**
> >
> > PBS-Attn improves performance by grouping scattered key tokens with similar behaviors (e.g., "heavy hitters") into contiguous blocks. This makes the blocks more distinct (i.e., highly dense vs. mostly empty), which significantly simplifies the task for the block selection algorithm.
> > **The clearest evidence is the direct comparison between PBS-Attn and the MeanPooling baseline. Both methods use the exact same block selection algorithm (MeanPooling), yet PBS-Attn achieves substantial gains solely due to the permuted structure:**
> >
> > - **+4.54** average score on LongBench (Table 1, Qwen-2.5-7B-1M)
> > - **+8.15** average score on LongBenchv2 (Table 2, Qwen-2.5-7B-1M)
> >
> > This advantage also extends to comparisons with other sparse attention methods: **at a given efficiency level (accounting for block sparsity, overhead, etc.), PBS-Attn selects tokens that account for a higher proportion of the total attention mass. This higher coverage directly translates into the improved accuracy observed in our benchmarks.**
> >
> > We have incorporated a brief analysis reflecting this insight into the revision. Thank you again for your valuable feedback!

---

> > > ### Author Response · Authors · 2025-11-27
> > >
> > > > **W4:** The description of how permutations are implemented in memory is vague. If permutation involves physically reordering the K/V cache or attention tensors, the memory movement could dominate runtime, negating theoretical speedups. The paper should explicitly clarify whether permutations are handled via index mapping (logical reordering) or physical memory rearrangement. As attention is typically memory-bound, even small data shuffling can be expensive.
> > > >
> > >
> > > **A4:**
> > >
> > > This is a very insightful comment. **To properly address the memory overhead and its influence, we have expanded the analysis in the permutation overhead section (Section D, Appendix) of the revision.**
> > >
> > > Our current sparse attention kernel implementation operates on **physically permuted tensors**, where we replace the original tensor with the permuted one. This process is implemented via a `torch.gather` call, which creates a temporary buffer equivalent to the size of the original tensor (approximately 32MiB per head for Llama-3.1-8B at 128K context), which is not significant. We believe it is feasible to eliminate this overhead with further kernel optimization by using logical reordering and directly loading scattered tokens into the kernel via index remapping. However, as the performance effect of loading non-consecutive elements is unclear, we view this as a promising avenue for future optimization.
> > >
> > > Theoretically, **the prefilling phase of LLMs is in fact compute-bound**, as it involves a massive amount of parallel computation, whereas the decoding phase is memory-bound due to the need to load the entire KV cache at each step. Therefore, during the prefilling phase, the insignificant memory movement required for permutation is unlikely to be the bottleneck for speedup.
> > >
> > > Empirically, **our evaluations show that memory movement is indeed not a bottleneck for overall speedup**. All our efficiency evaluations are empirical and demonstrate real speedups, which are achieved with our current permutation implementation. Furthermore, **Table 4** demonstrates that as context length increases, permutation time scales linearly, aligning with our analysis. The overhead ratio (including memory movement) decreases as the context length increases, dropping to just **3.1%** at a memory-intensive 512K length. This is further confirmed by our efficiency evaluation for a larger model (14B); even in this more memory-intensive scenario, the overall speedup trend remains consistent with the smaller 8B model (**Figure 13, Section F**).
> > >
> > > For more details, please refer to the revision of the paper. We appreciate your effort to improve our paper.

---

> > > > ### Author Response · Authors · 2025-11-27
> > > >
> > > > > **W5:** Despite the formal theorem, the proposed method largely appears as an engineering improvement to block-sparse attention rather than a conceptual advance in understanding attention sparsity. There is little theoretical or empirical insight into how permutation interacts with attention distribution, context structure, or causal masking beyond its computational benefit.
> > > > >
> > > >
> > > > **A5:**
> > > >
> > > > Thank you for the comment. We respectfully disagree with the characterization of the method as solely an engineering improvement. We believe PBS-Attn offers a conceptual advance by revealing that **structural order is a fundamental, yet overlooked, dimension of attention sparsity.** As discussed in **A1**, we have significantly expanded the method section (**Section 3.3**) and added an in-depth analysis of the permutation effect on block sparsity (**Section B**).
> > > >
> > > > Existing literature largely treats the attention matrix as a static object where one must find the best *selection* algorithm. Our work demonstrates that the matrix structure itself is mutable. We show that **scattered** "heavy hitter" tokens fundamentally limit the efficiency of *any* block-based selection method. By introducing permutation, we transform the problem from "selecting scattered blocks" to "densifying information," proving that **restructuring the attention matrix can serve as a distinct axis for optimizing block-sparse attention.**
> > > >
> > > > Our new analysis in **Section B** proves that the efficiency gain stems specifically from clustering "heavy hitter" tokens, grounded by a formal coverage definition (**Eq. 16**) and statistical analysis. The newly introduced baselines demonstrate that clustering via global information yields greater benefits than using local information, and that the method is robust to the selection of the query subset used for scoring. The results also confirm that the attention distribution is long-tailed, and clustering tokens from the tail of the distribution can significantly reduce the number of blocks that need to be retrieved (**Figure 7**).
> > > >
> > > > We also introduced **Segmented Permutation(Section 3.2)** specifically to reconcile the conflict between permutation (which is order-invariant) and causal masking (which is order-strict). This is not just an engineering fix but a novel formulation that allows global reordering optimizations to coexist with autoregressive constraints, addressing a theoretical gap in applying permutation to modern causal LLMs.
> > > >
> > > > In summary, PBS-Attn advances the understanding of attention sparsity by proving that **optimal block sparsity requires both selecting important blocks AND restructuring the sequence to maximize block information density.**

---

### Official Review · Reviewer_NZsM · 2025-10-26

**Soundness:** 3
**Presentation:** 3
**Contribution:** 3
**Rating:** 6
**Confidence:** 3

**Summary:**

The paper addresses the computational scaling issue of self-attention in large language models (LLMs) by proposing Permuted Block-Sparse Attention (PBS-Attn). The core idea is to leverage the permutation invariance property of the attention mechanism, reordering query or key tokens via segmented permutations to better align attention patterns with block-sparse structures. This method aims to significantly increase block-level sparsity and, consequently, improve memory and speed efficiency during long-context LLM inference, particularly in the prefilling stage. Comprehensive experiments on LongBench and LongBenchv2 benchmarks show PBS-Attn achieves competitive or superior accuracy relative to strong block-sparse baselines with notable speedups.

**Strengths:**

1. The paper presents a well-motivated idea grounded in a clear theoretical foundation.
2. The segmented permutation framework is plug-and-play and agnostic to the block selection algorithm. The approach is modular, supporting extensions and integration with existing block-sparse attention methods
3. PBS-Attn achieves near-full-attention accuracy with substantial runtime savings, outperforming recent baselines such as FlexPrefill and XAttention.

**Weaknesses:**

1. The method currently targets only the prefill stage. Its applicability to decoding or training phases is not explored.
2. The paper asserts “minimal performance degradation,” but from Table 1 and Table 2, there are domains or tasks (e.g., Qwen-2.5-7B-1M on LongBench, Code and Few-Shot Learning categories) where PBS-Attn performs slightly below the full attention baseline. No qualitative or error analyses are provided to identify failure modes or classes of inputs for which the approach may underperform.
3. Figure 2 and the supplementary distinctly show the ability of permutation to focus major attention mass into sparse bands, but there are no breakdowns of how different heads, layers affect the effectiveness of the permutation. For example, does the benefit hold in higher layers or only in early ones? A more systematic visualization set, perhaps showing variance across attention heads or real-world text types, is needed to understand the boundary of efficacy.
4. How sensitive is the approach to block size ($B$), segment size ($S$), and selection threshold? Are there hyperparameter settings for which the performance or efficiency gain vanishes or reverses?
5. What are the scaling laws of PBS-Attn efficiency as model size increases? Are there performance cliffs where the overhead consumes speedup?
6. Can the authors analytically bound or provide intuition or theory for the maximum achievable block sparsity benefit achievable by their segmented permutation approach (Section 3.2, Figure 4), or is the effect purely empirical?
7. Finally, it would be valuable to clarify whether the segmented permutation could, in extreme cases, lead to information leakage or violations of causal masking.

**Questions:**

See the Weaknesses.

---

> ### Author Response · Authors · 2025-11-27
>
> We thank you for your insightful comments and deep engagement with our work. We will address your concerns below.
>
> > **W1:** The method currently targets only the prefill stage. Its applicability to decoding or training phases is not explored.
> >
>
> **A1:**
>
> Thank you for this comment. We would like to clarify that PBS-Attn is **specifically designed and optimized for the prefilling phase** of inference in a training-free manner. This focus aligns with well-established literature in this domain, such as Minference [1], FlexPrefill [2], and XAttention [3].
>
> **Regarding decoding applicability:**
>
> In fact, there is a fundamental distinction between accelerating prefilling and decoding: **prefilling is typically compute-bound, whereas decoding is memory (I/O)-bound**. Consequently, prefill acceleration methods (including PBS-Attn and the works listed above) aim to reduce computation while often keeping the KV cache size intact. In contrast, decoding acceleration methods, such as H2O [5] and SnapKV [6], focus on reducing the KV cache size to alleviate memory bottlenecks. Importantly, these approaches are often complementary and can be deployed together in scenarios like Prefill-Decoding (PD) disaggregation.
>
> Looking ahead, we believe **permutation holds significant potential for the decoding phase as well**. By grouping key tokens with similar behaviors, permutation could be combined with retrieval-based decoding methods like Quest [7] to reduce the retrieval of irrelevant keys, leading to further acceleration and sparsity. We view this as a promising direction for future work.
>
> **Regarding training applicability:**
> Our permutation method (Query-aware Key Permutation) is specifically tailored to exploit the stable "heavy hitter" patterns found in **pretrained LLMs**. We position PBS-Attn strictly as a **plug-and-play inference optimization** that requires no retraining, similar to existing block-sparse techniques.
>
> [1] MInference 1.0: Accelerating Pre-filling for Long-Context LLMs via Dynamic Sparse Attention
>
> [2] FlexPrefill: A Context-Aware Sparse Attention Mechanism for Efficient Long-Sequence Inference
>
> [3] XAttention: Block Sparse Attention with Antidiagonal Scoring
>
> [4] SeerAttention: Learning Intrinsic Sparse Attention in Your LLMs
>
> [5] H2O: Heavy-Hitter Oracle for Efficient Generative Inference of Large Language Models
>
> [6] SnapKV: LLM Knows What You are Looking for Before Generation
>
> [7] Quest: Query-Aware Sparsity for Efficient Long-Context LLM Inference
>
> > **W2:** The paper asserts “minimal performance degradation,” but from Table 1 and Table 2, there are domains or tasks (e.g., Qwen-2.5-7B-1M on LongBench, Code and Few-Shot Learning categories) where PBS-Attn performs slightly below the full attention baseline. No qualitative or error analyses are provided to identify failure modes or classes of inputs for which the approach may underperform.
> >
>
> **A2:**
>
> Thank you for this constructive suggestion.
>
> We would like to clarify that for the evaluation of sparse attention methods, the benchmark result is not the only axis. What we fundamentally compare is the output quality at certain efficiency gains, which is why we plot trade-off figures (Figure 5) for analysis and ablation. The benchmark scores can be controlled via hyperparameters (e.g., the selection threshold); with higher attention coverage, the benchmark results improve. When coverage is sufficiently high, marginal result variations are often noise and non-trivial to analyze definitively.
>
> However, we can characterize typical failure modes when attention coverage is insufficient. This behavior applies to all sparse methods, not just PBS-Attn:
>
> - **Very low coverage:** The model output tends to degenerate into gibberish.
> - **Relatively high (but imperfect) coverage:** A minority of heads suffer from coverage degradation, the model tends to refuse to answer questions or retrieves (hallucinate) the wrong answer.
>
> Since the permutation effect is the core contribution of this paper, we believe analyzing failure modes at the **attention head level** provides more insight than looking at specific input classes. In **Appendix B** of the revision, we have provided a detailed head-level failure mode analysis. As shown in **Figure 9**, for heads with specific patterns (e.g., "Slash Lines" or highly query-specific patterns), permutation can potentially make it harder for the selection algorithm to capture all critical tokens. This could be a potential cause of failure in certain scenarios.
>
> We hope this explanation makes sense to you.

---

> > ### Author Response · Authors · 2025-11-27
> >
> > > **W3:** Figure 2 and the supplementary distinctly show the ability of permutation to focus major attention mass into sparse bands, but there are no breakdowns of how different heads, layers affect the effectiveness of the permutation. For example, does the benefit hold in higher layers or only in early ones? A more systematic visualization set, perhaps showing variance across attention heads or real-world text types, is needed to understand the boundary of efficacy.
> > >
> >
> > **A3:**
> >
> > Thank you for this constructive suggestion. We agree that the original draft could benefit from a more systematic analysis of permutation behaviors across different layers and heads. To address this, **we have added a comprehensive evaluation in Appendix B of the revision, analyzing the effect of permutation on block-level sparsity across three axes: method-wise, layer-wise, and head-wise.**
> >
> > We begin by formally defining block-level attention coverage (Eq. 16) and evaluating performance on real-world text data. Specifically, we use Llama 3.1 8B with 16K inputs (the maximum feasible length for materializing the full oracle attention matrix within our memory budget).
> >
> > **Method-wise**, we compare the proposed Last-Block-Query-Attn permutation against four baselines: No Permutation, Random Permutation, Greedy-Query-aware Permutation, and Random-Query-Attn Permutation (please refer to the revision for detailed definitions). **Figure 6** illustrates the overall sparsity gain ranking: Last-Query-Attn  $\approx$ Random-Query-Attn > Greedy > No permutation > Random Permutation. This indicates that **the gain primarily stems from clustering globally important key tokens** (also known as “heavy hitter” tokens, see Figure 2 for this effect), as global scoring schemes yield higher sparsity than using local query centroids (Last-Query-Attn > Greedy). Furthermore, **global scoring proves insensitive to the specific query subset selection** (Last-Query-Attn  $\approx$ Random-Query-Attn), using the last block of queries is only a practical choice.
> >
> > **Layer-wise**, we evaluate sparsity gains at different attention coverage levels (**Figure 7**). The results imply that layers respond differently to permutation. **Layer 0 consistently exhibits significant sparsity gains across all coverage levels. For the remaining layers, the gain increases with the coverage level.** This aligns with the long-tailed nature of attention distribution(Figure 6), and the "heavy hitter" tokens can be scattered across the sequence. Permutation clusters them, meaning fewer blocks need to be retrieved to capture the same attention mass. Notably, **the middle-to-deep layers benefit the most from permutation**.
> >
> > **Head-wise**, we **select three representative layers to analyze individual heads according to Figure 7**: Layer 0 (consistent large margin improvement), Layer 14 (marginal improvement), and Layer 21 (improvement scales with coverage) (Figure 8). **In Layer 0, nearly all heads show sparsity gains, with some achieving substantial improvements. In Layers 14 and 21, the vast majority of heads show improvement, while a small minority show only marginal gains (e.g., Layer 21 Head 2) or even degradation (e.g., Layer 14 Head 26).**
> >
> > **We provide a failure mode analysis for these cases in Figure 9**. We find that **for highly query-specific heads (where different queries attend to distinct sets of keys), the sparsity gain is marginal. In addition, for heads dominated by the "Slash Line" pattern (where queries attend to keys at fixed intervals), permutation can degrade sparsity.** This occurs because selecting diagonal blocks is naturally the optimal strategy for "Slash Line" patterns, and any permutation disrupts this structure. Despite these few failure cases, overall sparsity improves significantly. Performance could be further enhanced by a pruning strategy that avoids permuting these specific heads, which we leave for future work.
> >
> > For more details, please refer to Section B in the Appendix. Thank you again for helping us improve our paper!

---

> > > ### Author Response · Authors · 2025-11-27
> > >
> > > > **W4:** How sensitive is the approach to block size (), segment size (), and selection threshold? Are there hyperparameter settings for which the performance or efficiency gain vanishes or reverses?
> > > >
> > >
> > > **A4:**
> > >
> > > Thank you for the question.
> > >
> > > We have ablated both block size and segment size in **Section 4.3** of the revision by plotting the LongBench score-density trade-off figures (**Figure 5**). Note that the x-axis (density) is directly controlled by the selection threshold.
> > >
> > > Regarding the sensitivities to these parameters
> > >
> > > - Segment size ($S$) controls the permutation interval. Larger $S$ allows higher sparsity gain from permutation but introduces more diagonal segment computations. Given this trade-off, performance with respect to $S$ is relatively robust (**Figure 5(b)**).
> > > - Block size ($B$)controls the granularity for block selection. An overly large $B$ could diminish the permutation effect and include redundant tokens in the selection. Performance aligns closely at most density levels for $B \le 128$. We use $B=128$, in line with previously established methods like FlexPrefill and XAttention.
> > > - The selection threshold directly controls the sparsity of sparse attention; a low selection threshold will lead to insufficient attention coverage and performance degradation (**Figure 5(a)-(c)**). Performance is naturally sensitive to this parameter, as is true for all sparse attention methods.
> > >
> > > At a certain sparsity level (controlled by the selection threshold), **the effect of segment size and block size is not significant**. However**, an overly small selection threshold leads to performance degradation due to low attention coverage, while an overly large threshold causes efficiency gains to vanish due to the redundancy of token selection** (with a density of 1, the method falls back to full attention).
> > >
> > > > **W5:** What are the scaling laws of PBS-Attn efficiency as model size increases? Are there performance cliffs where the overhead consumes speedup?
> > > >
> > >
> > > **A5:**
> > >
> > > Thank you for the question. Evaluating larger long-context models is extremely expensive. We made our best effort and added the evaluation on a 14B model (Qwen-2.5-14B-1M) to validate the effectiveness of PBS-Attn on larger models.
> > >
> > > We added a new section, Section F, in the Appendix of the revision evaluating Qwen-2.5-14B-1M on LongBench as well as its efficiency. **Table 6(shown below) demonstrates that PBS-Attn consistently outperforms the baselines on this larger LLM, demonstrating scalability in performance.** We further validate its **efficiency gain in Figure 13.** Compared to the speedup of the 7B model in Figure 3, PBS-Attn achieves a similar speedup trend, where it has a nearly 2x speedup at a context length of 128K (maximum evaluated length without tensor parallelism); this **matches the results of the 7B model, indicating the scalability of PBS-Attn on efficiency.**
> > >
> > > | Method | Single-Doc QA | Multi-Doc QA | Summarization | Few-shot Learning | Code | Synthetic | Avg. |
> > > | --- | --- | --- | --- | --- | --- | --- | --- |
> > > | Full | 47.33 | 47.44 | 15.55 | 59.13 | 18.67 | 71.33 | 43.24 |
> > > | MInference | 46.50 | 45.73 | 15.56 | 57.23 | 18.76 | 63.33 | 41.19 |
> > > | FlexPrefill | 44.73 | 43.37 | 15.62 | 53.55 | 9.88 | 35.00 | 33.69 |
> > > | XAttention | **46.65** | 46.03 | **15.63** | **58.69** | **19.56** | 63.33 | 41.65 |
> > > | MeanPooling | 45.41 | 45.57 | 15.58 | 57.40 | 17.25 | 35.56 | 36.13 |
> > > | **PBS-Attn** | 46.56 | **46.43** | 15.48 | 58.51 | 17.53 | **67.17** | **41.95** |
> > >
> > > > **W6:** Can the authors analytically bound or provide intuition or theory for the maximum achievable block sparsity benefit achievable by their segmented permutation approach (Section 3.2, Figure 4), or is the effect purely empirical?
> > > >
> > >
> > >
> > > **A6:**
> > >
> > > Thank you for the question. We can indeed provide an analytical intuition for the maximum sparsity benefit based on the "heavy hitter" nature of attention.
> > >
> > > Consider a single off-diagonal segment of size $S$ divided into $K = S/B$ blocks.
> > >
> > > 1.  **Baseline (Scattered):** In the worst-case scenario for standard block-sparse attention, the critical "heavy hitter" tokens are scattered uniformly such that every block in the segment contains at least one important token. This forces the retrieval of all $K = S/B$ blocks.
> > > 2.  **PBS-Attn (Clustered):** Our permutation strategy sorts tokens by importance. Ideally, this packs all critical tokens into the very first block of the segment. This allows the model to retrieve just **1** block to cover the same attention mass.
> > >
> > > Therefore, the local sparsity gain (reduction in retrieved blocks) is analytically bounded by the ratio:
> > > $$ \text{Gain} \le \frac{S/B}{1} = \frac{S}{B} $$
> > >
> > > In our experiments ($S=256, B=128$), this gives a theoretical upper bound of a $2\times$ reduction in blocks computed for off-diagonal segments.

---

> > > > ### Author Response · Authors · 2025-11-27
> > > >
> > > > > **W7:** Finally, it would be valuable to clarify whether the segmented permutation could, in extreme cases, lead to information leakage or violations of causal masking.
> > > > >
> > > >
> > > > **A7:**
> > > >
> > > > We can confirm that segmented permutation **cannot** lead to information leakage or causal violations, by design. The guarantee comes from how we handle causality at two different levels:
> > > >
> > > > 1. **Inter-segment Causality (Guaranteed by Segmentation):**
> > > > The sequence is split into temporal segments (e.g., Segment 1 contains tokens $0 \dots S$, Segment 2 contains $S+1 \dots 2S$). Because permutation is strictly **intra-segment** (tokens are only shuffled *within* their own segment), any token in Segment 2 remains historically "after" every token in Segment 1, regardless of the internal order of Segment 1. Therefore, when Segment 2 attends to Segment 1, it is strictly attending to the past, preserving the autoregressive property.
> > > > 2. **Intra-segment Causality (Guaranteed by Diagonal Handling):**
> > > > The potential risk lies only within the "diagonal" segments (where a segment attends to itself). As noted in **Section 3.2**, our method explicitly identifies these on-diagonal segments. We treat these segments specially by ensuring the causal mask is applied based on the *original* token positions (or by computing them as standard dense causal attention blocks). This ensures that even within a segment, a token $t$ can never attend to $t+k$.
> > > >
> > > > Thus, there are no edge cases where future information leaks into the past.

---

### Official Review · Reviewer_sTLr · 2025-10-26

**Soundness:** 3
**Presentation:** 3
**Contribution:** 3
**Rating:** 6
**Confidence:** 4

**Summary:**

This paper introduces Permuted Block-Sparse Attention (PBS-Attn), a method designed to improve the computational efficiency of long-context large language models. The core idea is to leverage the permutation invariance of attention mechanisms to reorder query and key sequences in a way that enhances block-level sparsity, thus reducing redundant computation during the prefilling stage. The paper proposes a segmented permutation strategy that maintains causal structure while reordering tokens within segments, and a query-aware key permutation that aligns important key tokens. The approach is implemented efficiently using custom kernels, achieving up to 2.75× speedup with minimal accuracy degradation across benchmarks such as LongBench and LongBenchv2.

**Strengths:**

1. Novel idea with solid theory: The paper introduces a new optimization axis for sparse attention: token permutation. It builds on formal proofs of permutation invariance in attention, making the approach conceptually sound and mathematically rigorous.
2. Strong empirical results: PBS-Attn achieves up to 2.75× speedup with minimal accuracy loss, showing consistent gains across two major long-context LLMs and benchmarks.
3. Orthogonal contribution: The method complements existing block-selection techniques, providing a new dimension for improving block sparsity without modifying model architecture.

**Weaknesses:**

1. Incomplete evaluation: Missing key long-context benchmarks such as InfiniteBench and RULER, which limits understanding of the method’s scalability and robustness across diverse context lengths.
2. Unclear GQA handling: It remains unclear whether GQA heads share the same permutation pattern or whether the permutation is based on query heads rather than key-value heads.
3. Limited generality of the scoring method: The query-aware key permutation relies on the final queries, which may not perform well in chunk prefill or multi-round scenarios.
4. Prefill-only applicability: The method currently accelerates only the prefilling stage; its potential for decoding remains unexplored.
5. Insufficient overhead analysis: The runtime and memory cost of computing and applying permutations are not deeply quantified, making it difficult to assess net efficiency gains at scale.

**Questions:**

1. How are permutation patterns handled under GQA—do all heads in a group share one pattern?
2. Could alternative permutation scoring schemes improve robustness beyond final-query reliance?
3. Is there a feasible way to extend PBS-Attn to decoding or streaming inference?
4. What is the measured permutation overhead relative to overall speedup, especially beyond 512K tokens?
5. What is the effect of block size in addition to the Effect of Segment Size?

---

> ### Author Response · Authors · 2025-11-27
>
> We are grateful for your comprehensive evaluation and valuable suggestions. We will address your concerns below.
>
> > **W1:** Incomplete evaluation: Missing key long-context benchmarks such as InfiniteBench and RULER, which limits understanding of the method’s scalability and robustness across diverse context lengths.
> >
>
> **A1:**
>
> Thank you for your constructive suggestion. We agree that the paper could be further enhanced with more comprehensive evaluations. While computational constraints make it infeasible to run every available benchmark, we have strategically selected **RULER** to address your concerns regarding robustness across context lengths. We believe that **RULER** (for systematic length probing) combined with our existing **LongBenchv2** (which covers real-world tasks up to 2M tokens) creates a rigorous evaluation suite that effectively covers the task types InfiniteBench.
>
> We have added the RULER evaluations for **Llama-3.1-8B** and **Qwen-2.5-7B-1M** in **Table 3**(also shown below) of the revision.
>
> | Method | Llama 4K | Llama 8K | Llama 16K | Llama 32K | Llama 64K | Llama 128K | Llama Avg | Qwen 4K | Qwen 8K | Qwen 16K | Qwen 32K | Qwen 64K | Qwen 128K | Qwen Avg |
> | --- | --- | --- | --- | --- | --- | --- | --- | --- | --- | --- | --- | --- | --- | --- |
> | Full | 96.14 | 94.24 | 92.19 | 86.06 | 84.60 | 75.30 | 88.09 | 95.34 | 92.45 | 93.49 | 89.06 | 84.73 | 74.23 | 88.22 |
> | Minference | **95.98** | 93.67 | **91.95** | 85.55 | **83.48** | 70.47 | 86.85 | 94.01 | 91.30 | 91.60 | 89.09 | 81.30 | 70.10 | 86.23 |
> | FlexPrefill | 92.87 | 92.99 | 91.35 | 84.91 | 82.62 | 71.07 | 85.97 | 84.17 | 87.74 | 86.73 | 84.21 | 78.15 | 61.66 | 80.44 |
> | XAttention | 95.63 | **93.95** | 91.63 | 86.32 | 80.54 | 70.68 | 86.46 | 93.69 | 92.10 | 91.50 | 88.35 | 81.26 | 73.05 | 86.66 |
> | MeanPooling | 94.15 | 92.72 | 89.94 | 83.95 | 76.46 | 59.32 | 82.76 | 90.15 | 87.43 | 86.38 | 82.70 | 78.86 | 67.51 | 82.17 |
> | **PBS-Attn** | 95.83 | 93.85 | 91.46 | 85.18 | 82.51 | 66.98 | 85.97 | 93.27 | 90.77 | 90.54 | 85.54 | 81.50 | 70.61 | 85.37 |
> | **PBS-Attn⁺** | 95.72 | 93.85 | 91.23 | **87.05** | 81.27 | **72.09** | **86.87** | **94.06** | **92.24** | **92.59** | **89.31** | **84.37** | **73.71** | **87.71** |
>
> Unlike real-world data where tokens share semantic context, RULER tasks (e.g., retrieving random UUIDs) are synthetic and semantically meaningless. Consequently, coarse-grained selection schemes like mean pooling with block attention degrade on such tasks. To address this, we included a variation, PBS-Attn+ which adopts the fine-grained antidiagonal scoring scheme from XAttention.
>
> The results demonstrate that **permutation provides a consistent advantage regardless of the selection metric**:
>
> - **PBS-Attn vs. MeanPooling:** PBS-Attn achieves an average score improvement of **+3.21** on Llama-3.1-8B, with gains reaching **+7.66 at 128K context**.
> - **PBS-Attn+ vs. XAttention:** PBS-Attn+ outperforms XAttention by **+1.41** on Llama-3.1-8B and **+1.05** on Qwen-2.5-7B-1M. Notably, on the Qwen model, PBS-Attn+ matches the full attention baseline within a negligible margin of **0.51**.
>
> The underlying insight of this consistent improvement is that **permutation enhances block-level sparsity by densifying scattered information (clustering critical tokens), thus selecting more critical tokens and covering more of the attention mass.** Even in challenging synthetic scenarios where signals are essentially random noise (e.g., UUIDs), permutation successfully clusters relevant tokens, allowing the selection algorithm(whether coarse or fine-grained) to capture more attention within the same block budget.
>
> These findings confirm that PBS-Attn is both scalable and robust, effectively enhancing information retention even in challenging synthetic retrieval scenarios.

---

> > ### Author Response · Authors · 2025-11-27
> >
> > > **W2:** Unclear GQA handling: It remains unclear whether GQA heads share the same permutation pattern or whether the permutation is based on query heads rather than key-value heads.
> > >
> >
> > **A2:**
> >
> > Thank you for pointing this out. We clarify that the evaluations in the main paper utilize a default strategy where we **replicate the keys and values within GQA groups, allowing them to have different permuted orders for each query head**. We chose this approach **initially to maximize sparsity gains**.
> >
> > **To evaluate the feasibility of sharing the permutation within GQA groups, we added a new section in the revision of the paper (Section E, Appendix).** Here, we compare the attention coverage-density trade-off between the two strategies. **Figure 12** demonstrates that sharing the permutation leads to only a marginal drop in coverage at equivalent density levels. We further validated this on LongBench (**Table 5, replicated as below**), which confirms that the shared permutation strategy yields performance comparable to the default approach. This suggests that sharing the permutation is a viable alternative for the practical deployment of PBS-Attn.
> >
> > | Method | Single-Doc QA | Multi-Doc QA | Summarization | Few-shot Learning | Code | Synthetic | Avg. |
> > | --- | --- | --- | --- | --- | --- | --- | --- |
> > | **PBS-Attn(Default)** | 48.00 | 42.09 | 17.72 | 28.36 | 24.25 | 63.80 | 37.37 |
> > | **PBS-Attn(Shared Permutation)** | 48.38 | 41.47 | 17.84 | 27.77 | 23.86 | 63.92 | 37.21 |
> >
> > > **W3:** Limited generality of the scoring method: The query-aware key permutation relies on the final queries, which may not perform well in chunk prefill or multi-round scenarios.
> > >
> >
> > **A3:**
> >
> > Thank you for this thoughtful comment regarding the practical deployment of PBS-Attn. We appreciate the opportunity to clarify that **the sparsity gains are largely insensitive to the specific subset of queries used for scoring**. Using the final queries was primarily a design choice for simplicity and implementation efficiency in standard prefill settings.
> >
> > To address this concern, we have added a detailed analysis of the permutation effect in **Appendix B** of the revision, where we evaluate coverage and sparsity gains across three axes: method-wise, layer-wise, and head-wise.
> >
> > In the method-wise part, we introduced a baseline method using a **random subset** of queries for scoring. As shown in **Figure 6**, the results are nearly identical to the "Last-Block-Query" permutation. **This implies that it is not strictly necessary to use the final queries of the sequence. In chunk prefill or multi-round scenarios, PBS-Attn can be easily adapted by simply using queries from the current chunk or conversation round for scoring.**
> >
> > **The underlying insight is that the sparsity gain primarily stems from clustering the scattered “heavy hitter” key tokens (globally important tokens, a behavior exhibited by a majority of heads), so an arbitrary subset of queries can serve as an effective proxy.**
> >
> > For more details, please refer to Section B in the Appendix.

---

> ### Author Response · Authors · 2025-11-27
>
> > **W4:** Prefill-only applicability: The method currently accelerates only the prefilling stage; its potential for decoding remains unexplored.
> >
>
> **A4:**
>
> Thank you for this comment. We would like to clarify that PBS-Attn is **specifically designed and optimized (e.g., the segmented permutation design) for the prefilling phase**. This focus aligns with other well-established literature in this domain, such as Minference [1], FlexPrefill [2], XAttention [3], and SeerAttention [4], etc.
>
> In fact, there is a fundamental distinction between accelerating prefilling and decoding: **prefilling is typically compute-bound, whereas decoding is memory (I/O)-bound**. Consequently, prefill acceleration methods (including PBS-Attn and the works listed above) aim to reduce computation while often keeping the KV cache size intact. In contrast, decoding acceleration methods, such as H2O [5] and SnapKV [6], focus on reducing the KV cache size to alleviate memory bottlenecks. Importantly, these approaches are often complementary and can be deployed together in scenarios like Prefill-Decoding (PD) disaggregation.
>
> Looking ahead, we believe **permutation holds significant potential for the decoding phase as well**. By grouping key tokens with similar behaviors, permutation could be combined with retrieval-based decoding methods like Quest [7] to reduce the retrieval of irrelevant keys, leading to further acceleration and sparsity. We view this as a promising direction for future work.
>
> [1] MInference 1.0: Accelerating Pre-filling for Long-Context LLMs via Dynamic Sparse Attention
>
> [2] FlexPrefill: A Context-Aware Sparse Attention Mechanism for Efficient Long-Sequence Inference
>
> [3] XAttention: Block Sparse Attention with Antidiagonal Scoring
>
> [4] SeerAttention: Learning Intrinsic Sparse Attention in Your LLMs
>
> [5] H2O: Heavy-Hitter Oracle for Efficient Generative Inference of Large Language Models
>
> [6] SnapKV: LLM Knows What You are Looking for Before Generation
>
> [7] Quest: Query-Aware Sparsity for Efficient Long-Context LLM Inference
>
> > **W5**: Insufficient overhead analysis: The runtime and memory cost of computing and applying permutations are not deeply quantified, making it difficult to assess net efficiency gains at scale.
> >
>
> **A5:**
>
> Thank you for your constructive suggestion. We agree that the paper can be further strengthened by a detailed overhead analysis. To this end, we have significantly expanded the overhead analysis section (**Section D, Appendix**) in the revision.
>
> **Table 4** (as shown below in **A9**) demonstrates that the overhead (including computing and applying permutations, as well as block selection) accounts for only a minority of the overall time as the context length scales. For example, at a length of 512K, the overhead accounts for just **3.1%** of the overall runtime, confirming the practicality of PBS-Attn for long contexts.
>
> Furthermore, we added a memory cost analysis in **Section D**. The memory cost for computing the permutation scores is $O(BN)$, which scales linearly with context length. For $B=128$ and $N=128\text{K}$, this amounts to approximately **32MiB per head**, which is negligible relative to the total activation memory required for long sequences. We also analyzed the memory cost related to the permutation implementation itself; our current approach of physically permuting tensors introduces a 32MiB temporary buffer for 8B models (also insignificant), which can be further mitigated by implementing index remapping during kernel data loading.
>
> In **Table 4**, we verify that memory overhead does not diminish the speedup, as the relative overhead ratio decreases even in memory-intensive long-context scenarios. **Figure 13** further demonstrates that for a larger model (14B), which consumes significantly more memory for activations, PBS-Attn delivers speedups similar to those observed for the 8B model. This indicates that the cost of memory movement is not a bottleneck for the overall speedup.
>
> For more details, please refer to Section D in the revision.
>
> > **Q1:** How are permutation patterns handled under GQA—do all heads in a group share one pattern?
> >
>
> **A6:**
>
> The default strategy replicates the keys and allows them to have different permuted orders to maximize sparsity. We further show that sharing one pattern only leads to minimal impact on the sparsity gain and performance. For model details, please refer to A2 and Section E in the Appendix of the revision of the paper.
>
> > **Q2:** Could alternative permutation scoring schemes improve robustness beyond final-query reliance?
> >
>
> **A7:**
>
> The permutation essentially groups globally important key tokens and an arbitrary subset of queries should be able to be used for scoring. The final query method is a choice of simplicity and efficiency. For more details, please refer to A3 and Section B in the Appendix of the revision of the paper.

---

> > ### Author Response · Authors · 2025-11-27
> >
> > > **Q3:** Is there a feasible way to extend PBS-Attn to decoding or streaming inference?
> > >
> >
> > **A8:** Please refer to A4.
> >
> > > **Q4:** What is the measured permutation overhead relative to overall speedup, especially beyond 512K tokens?
> > >
> >
> > **A9:**
> >
> > Thank you for this question. Regarding the permutation overhead, we have conducted detailed benchmarks across various context lengths. While memory constraints on our current experimental setup preclude benchmarking at extremely long contexts (e.g., 1M tokens), we have provided a comprehensive analysis of permutation overhead for context lengths ranging from **4K to 512K** in **Appendix D (Table 4)** of the revision. We can predict the overall overheads by observing the trend at lengths that we can benchmark.
> >
> > The results demonstrate that the permutation time scales **linearly** with input length, whereas FlashAttention computation scales **quadratically**. Consequently, the total overhead (permutation + block selection) decreases significantly as a percentage of total runtime as the context length increases, dropping to just **3.1% at 512K** when permuting keys (our main method)**.** By observing the trend at these lengths—where overhead decreases significantly as a percentage of total runtime—we can reliably predict that the overhead remains negligible at even longer context lengths (e.g., beyond 512K).
> >
> > | Length | FlashAttention | Permutation | Block Selection | Attention | Total | Overhead | Speedup |
> > | --- | --- | --- | --- | --- | --- | --- | --- |
> > | ***Key Permutation*** |  |  |  |  |  |  |  |
> > | **4K** | 0.54 | 0.72 | 0.84 | 0.53 | 2.09 | 74.6% | 0.26× |
> > | **8K** | 1.78 | 0.82 | 0.86 | 1.31 | 2.99 | 56.2% | 0.60× |
> > | **16K** | 6.67 | 1.02 | 0.62 | 4.08 | 5.71 | 28.7% | 1.17× |
> > | **32K** | 26.10 | 1.68 | 0.60 | 13.83 | 16.11 | 14.2% | 1.62× |
> > | **64K** | 106.84 | 3.24 | 1.24 | 42.44 | 46.92 | 9.5% | 2.28× |
> > | **128K** | 443.29 | 6.22 | 3.21 | 150.45 | 159.87 | 5.9% | 2.77× |
> > | **256K** | 1837.32 | 13.87 | 11.81 | 563.54 | 589.22 | 4.4% | 3.12× |
> > | **512K** | 7496.67 | 26.76 | 41.99 | 2128.16 | 2196.91 | 3.1% | 3.41× |
> > | ***Query Permutation*** |  |  |  |  |  |  |  |
> > | **4K** | 0.54 | 0.70 | 0.81 | 0.63 | 2.31 | 65.4% | 0.23× |
> > | **8K** | 1.78 | 0.82 | 0.81 | 1.67 | 3.30 | 49.4% | 0.54× |
> > | **16K** | 6.67 | 1.20 | 0.49 | 5.06 | 6.75 | 25.0% | 0.99× |
> > | **32K** | 26.10 | 1.98 | 0.59 | 17.81 | 20.38 | 12.6% | 1.28× |
> > | **64K** | 106.84 | 3.52 | 1.25 | 52.95 | 57.72 | 8.3% | 1.85× |
> > | **128K** | 443.29 | 7.10 | 3.23 | 181.51 | 191.85 | 5.4% | 2.31× |
> > | **256K** | 1837.32 | 13.82 | 11.80 | 597.04 | 622.66 | 4.1% | 2.95× |
> > | **512K** | 7496.67 | 26.83 | 41.91 | 2481.68 | 2550.42 | 2.7% | 2.94× |
> >
> > > **Q5:** What is the effect of block size in addition to the Effect of Segment Size?
> > >
> >
> > **A10:**
> >
> > Thank you for the question. We have added a new ablation study in the revision to analyze the effect of block size. In the expanded **Section 4.3**, we visualize the LongBench score-density trade-off in **Figure 5(c)** for block sizes of 64, 128, and 256.
> >
> > The results indicate that smaller blocks ($B=64$) provide finer granularity, yielding better performance at very low densities ($<0.15$) by minimizing redundancy. Conversely, larger blocks ($B=256$) suffer from rapid degradation at low density budgets, as coarse selection forces the inclusion of redundant non-critical tokens. An intermediate block size of **$B=128$** strikes the optimal balance between granularity and efficiency, which is why we adopt it for our main method.

---

### Official Review · Reviewer_qNdW · 2025-10-30

**Soundness:** 2
**Presentation:** 2
**Contribution:** 2
**Rating:** 2
**Confidence:** 4

**Summary:**

The paper proposes PBS-Attn, a new block-sparse attention mechanism that permutes queries, keys, and values to better aggregate high attention scores. By leveraging the permutation-invariant and -equivariant properties of queries and key-value pairs, PBS-Attn clusters high attention values more effectively, thereby increasing attention coverage under the same sparsity budget. Experiments compare PBS-Attn with various sparse attention baselines in terms of both performance and prefill latency.

**Strengths:**

1. **Clear formulation:** The formulation of the attention computation and the analysis of permutation properties are clearly presented.
2. **Competitive results:** PBS-Attn achieves competitive performance across four sparse attention baselines. Although it does not consistently achieve the best score on every benchmark, it attains the highest average performance overall.

**Weaknesses:**

1. **Relation to prior work:** The paper lacks sufficient discussion of its relation and distinction from previous research. The idea of using permutation to better aggregate sparse attention scores has been explored in prior works such as [1]. The authors are encouraged to highlight the key differences and contributions relative to these methods.
2. **Global importance score computation:** The rationale for using the last query block to compute global importance is not clearly explained. It remains unclear how well this last block represents the entire attention spectrum, and no statistical analysis is provided.
3. **Lack of descriptive algorithm explanation:** The core design of the permuted block-sparse attention is condensed into Algorithm 1’s pseudocode, but lacks a detailed descriptive explanation in the main text. This omission may hinder readers’ understanding of the method.

[1] Kitaev, Nikita, Łukasz Kaiser, and Anselm Levskaya. “Reformer: The Efficient Transformer.”

**Questions:**

1. What is the permutation overhead under moderate context lengths, e.g., 4K tokens?
2. What is the numerical improvement in attention coverage achieved through permutation?
3. Did the authors implement any custom CUDA kernels to achieve the reported speedups with PBS-Attn?

---

> ### Author Response · Authors · 2025-11-27
>
> > **W1: Relation to prior work:** The paper lacks sufficient discussion of its relation and distinction from previous research. The idea of using permutation to better aggregate sparse attention scores has been explored in prior works such as [1]. The authors are encouraged to highlight the key differences and contributions relative to these methods.
> >
>
> **A1:**
> We thank the reviewer for pointing out the connection to prior works utilizing permutation, specifically Reformer. In **Section 5** of the original draft, we primarily focused on sparse attention methods involving token permutation in the context of modern LLMs. **We have added a discussion of Reformer in the revised version.** Although both methods utilize the idea of token permutation, there are three distinct differences between PBS-Attn and Reformer:
>
> 1. PBS-Attn is designed for the post-training regime and serves as a plug-and-play approach for modern LLMs without additional training. In contrast, Reformer necessitates a "Shared-QK" architecture (where queries and keys are identical), which is not applicable to current LLMs without significant retraining and architectural changes.
> 2. PBS-Attn uses **Query-aware Key Permutation** to cluster dominant key tokens based on attention patterns, whereas Reformer relies on Locality-Sensitive Hashing (LSH) to group tokens. We evaluated LSH for permutation in our pilot experiments and found that it did not yield good results, suggesting it may not be suitable for pre-trained LLMs.
> 3. Naive permutation breaks the causal mask required by LLMs. While Reformer addresses this with complex masking logic, our work introduces **Segmented Permutation**. This novel structural constraint strictly preserves inter-segment causality while maximizing block-level sparsity within segments, making it uniquely optimized for the prefilling stage of long-context inference.

---

> > ### Author Response · Authors · 2025-11-27
> >
> > > **W2: Global importance score computation:** The rationale for using the last query block to compute global importance is not clearly explained. It remains unclear how well this last block represents the entire attention spectrum, and no statistical analysis is provided.
> > >
> >
> > **A2:**
> >
> > Thank you for this insightful comment. We agree that it is essential to properly explain the rationale behind the last-query-block-attention permutation strategy proposed in the paper.
> >
> > To address this, **we have added a comprehensive analysis in Appendix B (referenced in Section 3.3 of the main text), where we evaluate the effect of permutation across three axes: method-wise, layer-wise, and head-wise.** Specifically, we measure the block-level sparsity gain achieved with oracle selection on the permuted attention matrix.
> >
> > **Method-wise**, we introduce four baselines: No Permutation, Random Permutation, Greedy Query-aware Key Permutation, and Random-Query-Attention-based Key Permutation (please refer to the revision for detailed definitions). Figure 6 demonstrates that for the overall sparsity gain: Last-Query-Attn  $\approx$ Random-Query-Attn > Greedy > No permutation > Random Permutation
> >
> > This implies that:
> >
> > 1. **The sparsity gain from permutation primarily stems from clustering “heavy hitter” tokens** (defined in [1], also recognized as “vertical lines” in [2][3]), which represent critical tokens for all queries.
> > 2. **The sparsity gain is insensitive to the selection of queries used to calculate proxy scores**, as the Random-Query and Last-Block-Query strategies yielded nearly identical results. Consequently, using the last block of queries is validated as an efficient and practical choice.
> >
> > **Layer-wise**, we measure the sparsity gain at different coverage levels (Figure 7). For most layers, the sparsity gain increases as the coverage level increases (with the exception of Layer 0, which consistently shows significant gains). Notably, the middle-to-deep layers benefit the most.
> >
> > **Head-wise**, we analyze the sparsity gain of individual attention heads at representative layers (Figure 8). We observed that heads respond differently to permutation: while the majority show significant or moderate sparsity gains, a minority show marginal gain or even degradation. We performed a failure mode analysis by visualizing the attention maps of these outliers. This revealed that heads dominated by the “Slash line” pattern can degrade under permutation, while heads with highly query-specific patterns benefit only marginally. This makes sense because our permutation here mainly groups the globally important tokens.
> >
> > For detailed analysis and results, please refer to **Section B** in the Appendix of the revision. We have also updated **Section 3.3** of the main paper to better illustrate the rationale for our permutation strategy. Thank you again for pinning this out.
> >
> > [1] H2O: Heavy-Hitter Oracle for Efficient Generative Inference of Large Language Models
> > [2] MInference 1.0: Accelerating Pre-filling for Long-Context LLMs via Dynamic Sparse Attention
> > [3] FlexPrefill: A Context-Aware Sparse Attention Mechanism for Efficient Long-Sequence Inference
> >
> > > **W3: Lack of descriptive algorithm explanation:** The core design of the permuted block-sparse attention is condensed into Algorithm 1’s pseudocode, but lacks a detailed descriptive explanation in the main text. This omission may hinder readers’ understanding of the method.
> > >
> >
> > A3:
> >
> > We appreciate your suggestion and **have revised Section 3.4 with a more detailed explanation of Algorithm 1 to provide a clear, step-by-step walkthrough of the proposed mechanism.** The revised text now explicitly describes the initial permutation of inputs, the generation of the sparse mask, the conditional tiled computation, and the final inverse permutation. Please check the revised version of the paper.
> >
> >
> > > **Q1:** What is the permutation overhead under moderate context lengths, e.g., 4K tokens?
> > >
> >
> > A4:
> >
> > **We have provided the detailed permutation overhead benchmarks in Table 4, Appendix D, including results for the requested 4K context length.** At 4K tokens, the permutation overhead accounts for approximately 74.6% of the total attention time, which is non-negligible.
> >
> > As observed in **Figure 3**, **the overhead at moderate context lengths (e.g., ≤8K) prevents *any* sparse attention method from outperforming the highly optimized FlashAttention**. However, it is worth noting that **PBS-Attn remains the most efficient among the sparse approaches in this regime**. For comparison, baselines like Flexprefill and Minference only begin to show speedups over FlashAttention at 64K and 128K tokens, respectively. Consequently, for practical deployments, we recommend a hybrid strategy: utilizing FlashAttention for sequences under 8K and switching to PBS-Attn for longer contexts to maximize efficiency. It should be very easy to implement using a simple routing logic.

---

> > > ### Author Response · Authors · 2025-11-27
> > >
> > > > **Q2:** What is the numerical improvement in attention coverage achieved through permutation?
> > > >
> > >
> > > A5:
> > >
> > > **We have added a comprehensive numerical evaluation in Appendix B (specifically Figures 6 and 7) to quantify these improvements.** We formally define block-level attention coverage in Eq. 16 and measure the gains using Llama-3.1-8B with a 16K context length.
> > >
> > > We chose 16K as it was the maximum feasible length for materializing the full oracle attention matrix within our memory budget. It is important to note that **inherent attention sparsity typically increases with context length**; therefore, the improvements reported here are likely conservative estimates, and the sparsity gains would be even more significant for longer contexts.
> > >
> > > We analyze the numerical improvement from two perspectives:
> > >
> > > 1. **Improvement in Attention Coverage:**
> > >
> > >     As shown in **Figure 6**, permutation increases absolute attention coverage by **1% to 2%** under the same block budget compared to the baseline. **This is actually a highly significant number in the context of the long-tailed attention distribution**. Gaining this specific coverage means the permutation successfully clusters scattered "heavy hitter" tokens that are otherwise extremely expensive to retrieve (i.e., they would usually require retrieving a vast number of additional blocks to capture without permutation).
> > >
> > > 2. **Improvement in Sparsity (Compute Savings):**
> > >
> > >     A more intuitive way to measure this impact is to look at the sparsity gain at a target coverage level. **Figure 7** demonstrates that to achieve a coverage of 0.975, permutation allows us to drop significantly more blocks. Most layers exhibit an **absolute sparsity gain of 5% to 20%**. This translates directly into computational efficiency, as the model can achieve the same retrieval quality while processing significantly fewer blocks.
> > >
> > >
> > > > **Q3:** Did the authors implement any custom CUDA kernels to achieve the reported speedups with PBS-Attn?
> > > >
> > >
> > > A6: We implemented custom kernels via Triton as we mentioned in Section 4.1, and the reported speedups are measured using these kernels.
> > >
> > > Here is the kernel implementation: https://anonymous.4open.science/r/pbs-attn-BB66/pbs_attn/src/kernels/permuted_block_sparse_attention.py
> > >
> > > And the efficiency benchmark code: https://anonymous.4open.science/r/pbs-attn-BB66/eval/efficiency/eval_efficiency.py

---

### Author Response · Authors · 2025-11-27
**General Response to Reviewers**

We thank the reviewers for their time and their insightful, constructive feedback. We are encouraged that the reviewers found our work:

1. Novel (**`sTLr`**)
2. Backed with solid theory (**`sTLr`**, **`NZsM`**, **`FuSK`**)
3. Delivering competitive empirical results (**`qNdW`**, **`sTLr`**, **`NZsM`**)

We value every thoughtful suggestion from the reviewers and have made tremendous efforts to revise our draft. Below is a summary of the major updates included in the revision:

- **In-depth analysis of the permutation effect on block-level sparsity (Section 3.3 and Section B, Appendix).**
    - Analyzed block attention coverage with a formal definition (Eq. 16) and statistical evaluations across three axes: method-wise (Figure 6), layer-wise (Figure 7), and head-wise (Figure 8). We also included a failure mode analysis (Figure 9).
    - Addressing concerns from: **`qNdW`** (W2, Q2), **`sTLr`** (W3, Q2), **`NZsM`** (W2, W3), **`FuSK`** (W1, W2, W5).

- **Evaluation on RULER (Section 4).**
    - Conducted an evaluation on the RULER benchmark to systematically assess the effectiveness of PBS-Attn (Table 3).
    - Addressing concerns from: **`sTLr`** (W1).

- **Evaluation on a larger LLM (Section F, Appendix).**
    - Added performance evaluation on a larger long-context LLM (Qwen-2.5-14B-1M) using LongBench (Table 6) and efficiency evaluation across various context lengths (Figure 13).
    - Addressing concerns from: **`NZsM`** (W5).

- **Detailed speed/memory overhead analysis (Section D, Appendix).**
    - Included detailed overhead benchmarking (Table 4) and an analysis of the memory cost regarding scoring and permutation.
    - Addressing concerns from: **`qNdW`** (Q1), **`sTLr`** (W5, Q4), **`FuSK`** (W4).

- **Ablation on block size (Section 4.3).**
    - Added ablation studies and analysis of the effect of block size (Figure 5(c)).
    - Addressing concerns from: **`qNdW`** (Q5), **`NZsM`** (W4).

- **Ablation on GQA handling (Section E, Appendix).**
    - Added coverage-density trade-offs with different GQA handling strategies (Figure 12) and actual performance evaluation on LongBench (Table 5).
    - Addressing concerns from: **`sTLr`** (W2, Q1).

- **Clarifications on the algorithm (Section 3.4).**
    - Added a detailed algorithm walkthrough in the method section.
    - Addressing concerns from: **`qNdW`** (W3).

- **Expanded Discussion on Relation to Prior Work (Section 5).**
    - Added a discussion on prior permutation-based methods (specifically Reformer).
    - Addressing concerns from: **`qNdW`** (W1).

**Code Availability:** We provide the source code and custom kernels at: https://anonymous.4open.science/r/pbs-attn-BB66

The modifications in the revision are highlighted in green. We appreciate the efforts made by the reviewers, and we believe these updates substantially strengthen the paper. We have responded to each reviewer individually below with detailed answers and experimental results.

---

### Meta-Review · Area_Chair_j2FW · 2025-12-26

**Summary:**

This paper proposes PBS-Attn, a plug-and-play permuted block-sparse attention method to boost long-context LLM prefill efficiency by enhancing block sparsity via causal-preserving segmented permutations. While addressing a critical efficiency bottleneck with sound permutation invariance analysis, it fails to fully resolve reviewer concerns. Key reviewer concerns include: (1) Insufficient explanation of permutation’s sparsity improvement mechanism and its interaction with known attention patterns (Reviewer FuSK); (2) Incomplete discussion on differences from prior permutation-based works (e.g., Reformer) (Reviewer qNdW); (3) Unclear rationale and robustness of the "last query block" scoring strategy (Reviewer qNdW, FuSK); (4) Incomplete evaluation (missing benchmarks like RULER, lack of large-model analysis initially) (Reviewer sTLr, NZsM);

**Reviewer Scores:**

NA

---

### Decision · Program_Chairs · 2026-01-26

Reject